



# Technical note: Diagnostic efficiency – specific evaluation of model performance

Robin Schwemmle[1], Dominic Demand[1], Markus Weiler[1]

[1]University of Freiburg, Faculty of Environment and Natural Resources, Chair of Hydrology, Freiburg, Germany

*Correspondence to*: Robin Schwemmle (robin.schwemmle@hydrology.uni-freiburg.de)

**Abstract.** Better understanding of the reasons why hydrological model performance is "good" or "poor" represents a crucial part for meaningful model evaluation. However, current evaluation efforts are mostly based on aggregated efficiency measures such as Kling-Gupta Efficiency ($KGE$) or Nash-Sutcliffe Efficiency ($NSE$). These aggregated measures only distinguish between "good" and "poor" model performance. Especially in the case of a "poor" model performance it is important to

identify the different errors which may have caused such unsatisfying predictions. These errors may origin from the model parameters, the model structure, and/or the input data. In order to provide more insight, we define three types of errors which may be related to their origin: constant error (e.g. caused by consistent input data error such as precipitation), dynamic error (e.g. structural model errors such as a deficient storage routine) and timing error (e.g. caused by input data errors or deficient model routines/parameters). Based on these types of errors, we propose the novel Diagnostic Efficiency ($DE$) measure, which

accounts for the three error types. The disaggregation of $DE$ into its three metric terms can be visualized in a plain radial space using diagnostic polar plots. A major advantage of this visualization technique is that error contributions can be clearly differentiated. In order to provide a proof of concept, we first generated errors systematically by mimicking the three error types (i.e. simulations are surrogated by manipulating observations). By computing $DE$ and the related diagnostic polar plots for the mimicked errors, we could then supply evidence for the concept. Finally, we tested the applicability of our approach

for a modelling example. For a particular catchment, we compared streamflow simulations realized with different parameter sets to the observed streamflow. For this modelling example, the diagnostic polar plot suggests, that dynamic errors explain the model performance to a large extent. The proposed evaluation approach provides a diagnostic tool for model developers and model users and the diagnostic polar plot facilitates interpretation of the proposed performance measure.



## 1 Introduction

Performance metrics quantify hydrological model performance. They are employed for calibration and evaluation purposes. For these purposes, the Nash-Sutcliffe efficiency (*NSE*; Nash and Sutcliffe, 1970) and the Kling-Gupta efficiency (*KGE*; Gupta et al., 2009) are two commonly used performance metrics in hydrology (e.g. Newman et al., 2017;Towner et al., 2019). *NSE* and *KGE* measure the overall model performance can be measured by only a single numerical value within the range of minus infinity and one. A value close to one indicates a better model performance, whereas with increasing distance to one the model performance deteriorates. From this point of view, the model performance can only be assessed in terms of "good" or "poor". However, cases of poor model performance immediately lead to the following questions: Why is my model performance not satisfying? What could improve the model performance?

In order to answer such questions, Gupta et al. (2008) proposed an evaluation approach that includes diagnostic information. Such a diagnostic approach requires appropriate information. Considering only the overall metric values of *NSE* and *KGE* may not provide any further insights. Additionally, an in-depth analysis of *KGE* metric terms may provide more information on the causes of the model error (e.g. Towner et al., 2019). Although including the *KGE* metric terms may enrich model evaluation, due to their statistical nature the link to hydrological process is less clear. Current diagnostic approaches are either based on entropy-based measures (Pechlivanidis et al., 2010) or on process-based signatures (Yilmaz et al., 2008;Shafii et al., 2017). The latter one improves measuring the realism of hydrological processes by capturing them in hydrological signatures. These signatures represent a main element of a powerful diagnostic approach (Gupta et al., 2008).

Although the numerical value of the overall model performance is diagnostically not meaningful, the overall model performance determines whether diagnostic information will be valuable to the modeller or not. Usually, diagnostic information may only be useful if the overall model performance does not fulfil the modeller's requirements. It will then be cumbersome to select the appropriate signatures or measures which may answer the modeller's questions about the causes. Visualising evaluation results in a comprehensive way poses another challenge for diagnostically meaningful interpretation. Therefore, we see a high potential in compressing the complex error terms into one diagram simplifying the interpretation. In this study, we propose a specific model evaluation approach which contributes to existing diagnostic evaluation approaches and builds on existing approaches.

## 2 Methodology

### 2.1 Diagnostic efficiency

In general, the quality of observations should be verified before simulations and observations are compared against each other. Observations with insufficient accuracy should not be considered for model evaluation. Likewise, accuracy of initial and





boundary conditions should be inspected beforehand. Remaining errors in hydrological simulations may then be caused by the
following origins:

- model parameters (e.g. Wagener and Gupta, 2005)
- model structure (e.g. Clark et al., 2008;Clark et al., 2011)
- input data (e.g. Yatheendradas et al., 2008)
- uncertainties in observations (e.g. Coxon et al., 2015)
- initial and boundary conditions (e.g. Staudinger et al., 2019)

Thus, within our approach we focus on errors caused by model parameters, model structure and input data. In order to diagnose
the origin of the errors, we define three error types linking to model parameters, model structure and input data:

- Constant error may have its origin in the input data or the model parameters. For example, errors may be caused by
  consistent input data error or by inappropriate model parameters causing consistent overestimation/underestimation.
- Dynamic error may have its origin in the model structure or the model parameters. For example, structural model
  errors (e.g. deficient storage routine) or deficient model parameters (e.g. parameters of the storage routine) may cause
  dynamic errors.
- Timing error may have its origin in the input data, the model structure or the model parameters. The error may be
  caused, for example, by input data errors and/or deficient model routines/parameters.

In order to contribute to existing diagnostic evaluation approaches we introduce the diagnostic efficiency (*DE*; Eq. 1):

$$DE = 1 - \sqrt{\overline{B_{rel}}^2 + |B_{area}|^2 + (r-1)^2}, \tag{1}$$

where $\overline{B_{rel}}$ is a measure for constant error, $|B_{area}|$ for dynamic error, and *r* for timing error. Similar to *NSE* and *KGE*, *DE* ranges
from 1 to -∞. *DE* = 1 indicates perfect agreement between simulations and observations.

First, we introduce the three terms which define the *DE*. The first two terms $\overline{B_{rel}}$ and $|B_{area}|$ are based on the flow duration
curve (FDC). Since FDC-based signatures do not include information on temporal performance, we have added correlation (*r*)
as a third term. $\overline{B_{rel}}$ reflects the constant error and is represented by the arithmetic mean of the relative bias (Eq. 2):

$$\overline{B_{rel}} = \frac{1}{N} \sum_{i=0}^{i=1} B_{rel}(i), \tag{2}$$

*i* represents the exceedance probability, *N* the total number of data points and $B_{rel}$ is the relative bias of the simulated and
observed flow duration curve; $\overline{B_{rel}} = 0$ indicates no constant error; $\overline{B_{rel}} < 0$ indicates a negative bias; $\overline{B_{rel}} > 0$ indicates a
positive bias. The relative bias between the simulated and observed flow duration curve ($B_{rel}$) calculates as follows (Eq. 3):

$$B_{rel}(i) = \frac{Q_{sim}(i) - Q_{obs}(i)}{Q_{obs}(i)}, \tag{3}$$

$Q_{sim}$ is the simulated streamflow at exceedance probability *i* and $Q_{obs}$ the observed streamflow at exceedance probability *i*.





The dynamic error is described by the absolute area of the residual bias ($|B_{area}|$; Eq. 4):

$$|B_{area}| = \int_0^1 |B_{res}(i)| \, di, \tag{4}$$

where the residual bias $B_{res}$ is integrated over the entire domain of the flow duration curve. Combining Eq. (2) and Eq. (3) results in:

$$B_{res}(i) = B_{rel}(i) - \overline{B_{rel}}, \tag{5}$$

by subtracting $\overline{B_{rel}}$ we remove the constant error and the dynamic error remains. $|B_{area}| = 0$ indicates no dynamic error; $|B_{area}| > 0$ indicates a dynamic error.

To consider timing errors, the Pearson's correlation coefficient ($r$) is calculated (Eq. 6):

$$r = \frac{\sum_{i=1}^n (Q_{obs}(i) - \mu_{obs})(Q_{sim}(i) - \mu_{sim})}{\sqrt{(\sum_{i=1}^n (Q_{obs}(i) - \mu_{obs})^2)(\sum_{i=1}^n (Q_{sim}(i) - \mu_{sim})^2)}}, \tag{6}$$

where $Q_{sim}$ is the simulated streamflow at time $t$, $Q_{obs}$ the observed streamflow at time $t$, $\mu_{obs}$ the simulated mean streamflow, and $\mu_{obs}$ the observed mean streamflow. Other non-parametric correlation measures could be used as well.

## 2.2 Diagnostic polar plot

$DE$ can be used as another aggregated efficiency by simply calculating the overall model performance. However, the aggregated value does only allow for a limited diagnosis since information of the metric terms is not interpreted. Thus, we project $DE$ and its metric terms in a radial plane (i.e. similar to a clock) to construct a diagnostic polar plot. An annotated version for a diagnostic polar plot is given in Fig. 3. For the diagnostic polar plot, we calculate the direction of the dynamic error ($B_{dir}$; Eq. 7):

$$B_{dir} = \int_0^{0.5} B_{res}(i) \, di, \tag{7}$$

where the integral of $B_{res}$ includes values from 0th percentile to 50th percentile. Since we removed the constant error (see Eq. 5), the left half of the integral is positive and the right half (i.e. 50th percentile to 100th percentile) will, thus, be negative and vice versa if the left half of the integral is negative.

In order to differentiate the dynamic error type, we computed the slope of the residual bias ($B_{slope}$; Eq. 8):

$$B_{slope} = \begin{cases} |B_{area}| \cdot (-1), & B_{dir} > 0 \\ |B_{area}|, & B_{dir} < 0 , \\ 0, & B_{dir} = 0 \end{cases} \tag{8}$$

$B_{slope} = 0$ expresses no dynamic error; $B_{slope} < 0$ indicates that there is a tendency of simulations to overestimate high flows and underestimate low flows while $B_{slope} > 0$ indicates a tendency of simulations to underestimate high flows and overestimate low flows.

We used the inverse tangent to derive the ratio between constant error and dynamic error in radians ($\varphi$; Eq. 9):

$$\varphi = arctan2(\overline{B_{rel}}, B_{slope}), \tag{9}$$





Instead of using a benchmark to decide whether model diagnostics is valuable or not, we introduce certain threshold for deviation-from-perfect. We set a threshold value ($l$) for which metric terms deviate from perfect and insert it in Eq. (1):

$$DE_l = 1 - \sqrt{l^2 + l^2 + ((1-l)-1)^2},$$
(11)

for this study $l$ is set by default to 0.05. Here, we assume that for a deficient simulation each metric term deviates at least 5% from its best value. $l$ can be either relaxed or expanded depending on the requirements of model accuracy. Correspondingly, $DE_l$ represents a threshold which discerns between a deficient simulation ($DE \leq DE_l$) and a good simulation ($DE > DE_l$). Finally, the following conditions describe whether a diagnosis can be drawn (Eq. 12):

$$Diagnosis = \begin{cases} yes, & |\overline{B_{rel}}| \leq l \ \& \ B_{slope} > l \ \& \ DE \leq DE_l \\ yes, & |\overline{B_{rel}}| > l \ \& \ B_{slope} \leq l \ \& \ DE \leq DE_l \ , \\ yes, & |\overline{B_{rel}}| > l \ \& \ B_{slope} > l \ \& \ DE \leq DE_l \end{cases}$$
(12)

There exists a special case for which timing error only can be diagnosed (Eq. 13):

$$Diagnosis = \ timing \ error \ only, \qquad |\overline{B_{rel}}| \leq l \ \& \ B_{slope} \leq l \ \& \ DE \leq DE_l,$$
(13)

If $DE$ and its metric terms are within the boundaries of acceptance, no diagnosis is required which is expressed by the following conditions (Eq. 14):

$$Diagnosis = \ no, \qquad |\overline{B_{rel}}| \leq l \ \& \ B_{slope} \leq l \ \& \ DE > DE_l,$$
(14)

In this case, the model performance is sufficiently accurate and can be denoted as a good simulation.

## 2.3 Comparison to *KGE* and *NSE*

In order to allow a comparison to commonly used *KGE* and *NSE*, we calculated the overall metric values and for *KGE* its three individual metric terms. We used the original *KGE* proposed by Gupta et al. (2009):

$$KGE = 1 - \sqrt{(\beta - 1)^2 + (\alpha - 1)^2 + (r - 1)^2},$$
(15)

where $\beta$ is the bias error, $\alpha$ represents the flow variability error, and $r$ shows the linear correlation between simulations and observations (Eq. 16):

$$KGE = 1 - \sqrt{\left(\frac{\mu_{sim}}{\mu_{obs}} - 1\right)^2 + \left(\frac{\sigma_{sim}}{\sigma_{obs}} - 1\right)^2 + (r - 1)^2},$$
(16)

where $\sigma_{obs}$ is the standard deviation in observations, $\sigma_{sim}$ the standard deviation in simulations. Moreover, we applied the polar plot concept (see Sect. 2.2) to *KGE* and the accompanying three metric terms.

*NSE* (Nash and Sutcliffe, 1970) calculates as follows (Eq. 17):

$$NSE = 1 - \frac{\sum_{t=1}^{t=T}(Q_{obs}(t) - Q_{sim}(t))^2}{\sum_{t=1}^{t=T}(Q_{obs}(t) - \mu_{obs})^2},$$
(17)





where $T$ is the total number of time steps, $Q_{sim}$ the simulated streamflow at time $t$, $Q_{obs}$ the observed streamflow at time $t$ and $\mu_{obs}$. NSE = 1 displays perfect fit between simulations and observations; NSE = 0 indicates that simulations performs equally well as the mean of the observations; NSE < 0 indicates that simulations perform worse than the mean of the observations.

**3 Proof of concept**

To provide a proof of concept any perennial streamflow time series coming from a near-natural catchment and having sufficiently long temporal record (i.e. > 30 years) may be used. We selected an observed streamflow time series from the CAMELS dataset (Fig. 1; Addor et al., 2017). In order to mimic specific model errors, we systematically manipulated the observed time series. Thus, we produced different time series which serve as a surrogate for simulated time series with a certain

error type which we call manipulated time series. These manipulated time series are characterised by a single error type or multiple error types, respectively. We calculated $DE$ for each manipulated time series and visualised the results in a diagnostic polar plot.

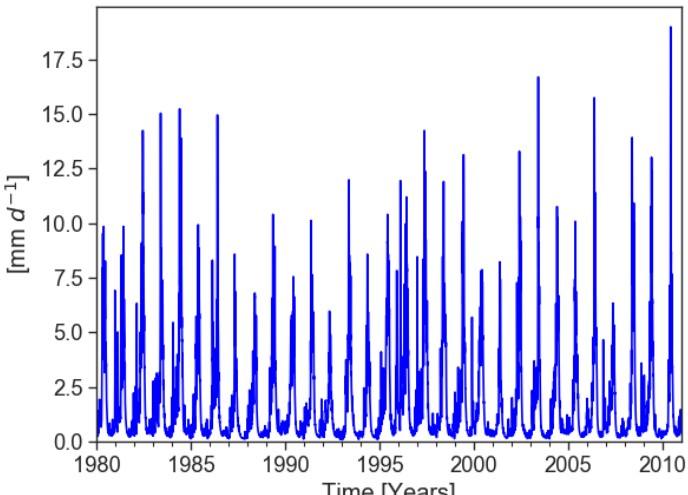

**Figure 1: Observed streamflow time series from CAMELS dataset (Addor et al., 2017; gauge_id: 13331500; gauge_name:**
**Minam River near Minam, OR, U.S.)**

**3.1 Mimicking errors**

In the following, we portray how we generated the manipulated time series mimicking modelling errors. Table 1 provides a brief summary on the error types and how we combined them. The resultant FDCs are illustrated in Figure 2. For the corresponding time series, we refer to the supplement (Fig. S1). We first describe the genesis of the time series for individual

errors:





(a) Positive constant error: We generated a positive offset by multiplying the observed time series with a constant 1.25 (see Fig. 2a and Fig. S1a). Constant requires to be > 1.

(b) Negative constant error: We generated a negative offset by multiplying the observed time series with a constant 0.75 (see Fig. 2b and Fig. S1b). Constant requires to be < 1.


(c) Positive dynamic error: We built a linearly interpolated vector $(1+p, …, 1, …, p)$ with $p$ set to 0.5. We then generated the error by multiplying the observed FDC with the linearly interpolated vector. With that, we increased high flows and decreased low flows. As a consequence, hydrological extremes are amplified (see Fig. 2c and Fig. S1c). Note that the original temporal order is maintained.

(d) Negative dynamic error: We built a linearly interpolated vector $(p, …, 1, …, 1+p)$ with $p$ set to 0.5. We then generated the error by multiplying the observed FDC with the linearly interpolated vector. With that, we decreased high flows and increased low flows. As a consequence, hydrological extremes are moderated (see Fig. 2d and Fig. S1d). Note that the original temporal order is maintained.


(e) We reproduced a timing error by randomizing the order of the observed time series (see Fig. 2e and Fig. S1e).

We then assembled the individual techniques (a-d) for the genesis of time series which are characterised by a combination of constant error and dynamic error. The two errors contribute with an equal share:


(f) Negative constant error and negative dynamic error (see Fig. 2f and Fig. S1f)

(g) Positive constant error and negative dynamic error (see Fig. 2g and Fig. S1g)

(h) Negative constant error and positive dynamic error (see Fig. 2h and Fig. S1h)

(i) Positive constant error and positive dynamic error (see Fig. 2i and Fig. S1i)


and time series which contain constant error, dynamic error (again both errors are contributing with an equal share) and timing error (a-e):

(j) Negative constant error, negative dynamic error and timing error (see Fig. S1j)

(k) Positive constant error, negative dynamic error and timing error (see Fig. S1k)

(l) Negative constant error, positive dynamic error and timing error (see Fig. S1l)


(m) Positive constant error, positive dynamic error and timing error (see Fig. S1m)

Note that for j-m FDCs are identical to f-i and are therefore not shown in Figure 2.

**Table 1: Summary on mimicked error types and its combinations as described in Sect. 3.1 (a-m). + (-) reflects a positive (negative) error type. For timing error, only one error type exists (x).**

|  | a | b | c | d | e | f | g | h | i | j | k | l | m |
|---|---|---|---|---|---|---|---|---|---|---|---|---|---|
| Constant error **(+/-)** | + | - |  |  |  | - | + | - | + | - | + | - | + |
| Dynamic error **(+/-)** |  |  | + | - |  | - | - | + | + | - | - | + | + |
| Timing error **(x)** |  |  |  |  | x |  |  |  |  | x | x | x | x |

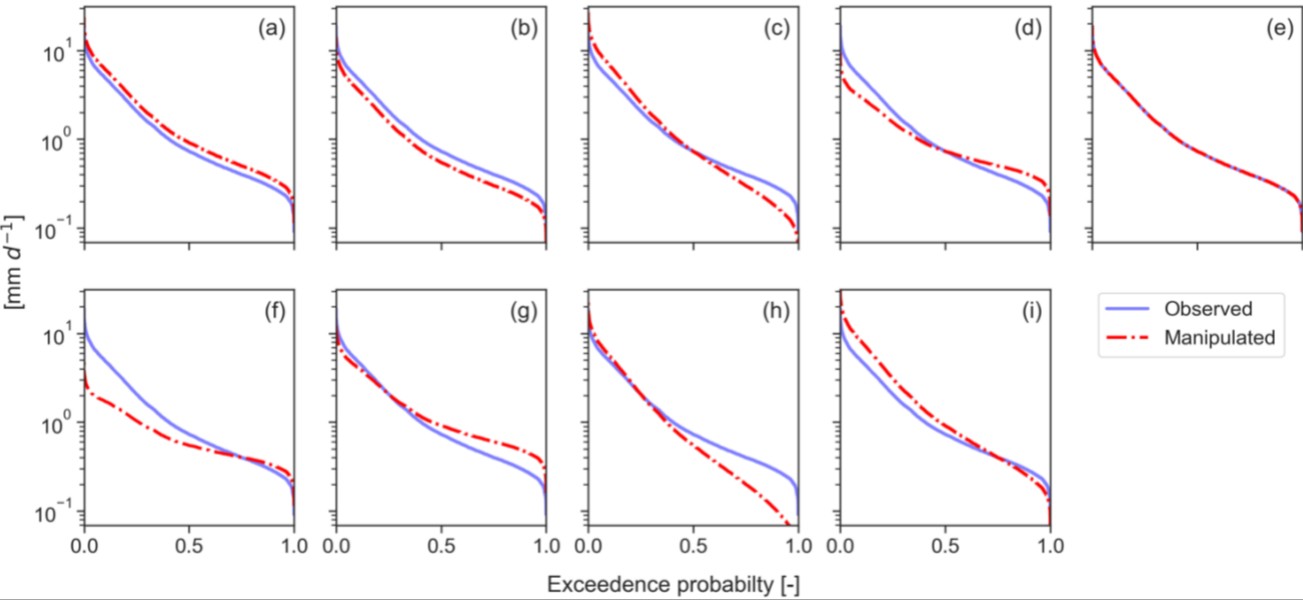


**Figure 2: Flow duration curves (FDCs) of observed (blue) and manipulated (dashed red) streamflow time series. Manipulated FDCs are depicted for (a-b) constant errors only, (c-d) dynamic errors only, (e) timing error only, and (f-i) combination of dynamic and constant errors. The combination of constant errors, dynamic errors and timing error is not shown, since their FDCs are identical to f-i. Y-axis is shown in log space.**

The diagnostic polar plot for mimicked error cases is shown in Fig. 3. Interdependently which error has been mimicked, related points are located in different error regions. For individual errors (a-d), related points are placed in the four cardinal directions of each region (Fig. 3). Within these regions the dominant error type can be easily identified. The more central the direction of the point, the more dominant is the error type. In case there is only a timing error present (e) an arrow with two ends instead of a point is used (Fig. 3). This is because dynamic error origin becomes arbitrary (i.e. high flows and low flows are being both

underestimated and overestimated (see Fig. S1e)). For combinations of constant and dynamic error (f-i), related points are located on boundaries of constant error and dynamic error meaning that both errors are equally dominant (Fig. 3). The same applies for combinations of constant error, dynamic error and timing error except that points shifted towards outer scope of the plot due to added timing error. Numeric values of $DE$ are listed in Table 2. $DE$ values are greater for individual errors (except for timing error) than for combined errors. Increasing the number of errors added to a time series, leads to lower $DE$.

For the numeric values of the individual metric terms, we refer to Table S1.

    A comparison of $DE$, $KGE$, and $NSE$ calculated for the manipulated time series is shown in Table 2. Numerically, $DE$ generally indicates a better performance than $KGE$ and $NSE$. Moreover, values for $DE$ exhibit a regular pattern (i.e. mimicking single error types or multiple error types, respectively, leads to an equidistant decrease in performance). By contrast, values for $KGE$ and $NSE$ are characterised by an irregular pattern (i.e. mimicking single error types or multiple error types, respectively, leads

to a non-equidistant decrease in performance). This non-equidistant decrease suggests that $KGE$ and $NSE$ are differently





sensitive to the mimicked errors. For example, lowest *KGE* values for single constant and dynamic errors are obtained by only introducing one error type (Table 2a-d). *NSE* is prone to timing errors (Table 2e), particularly to peak flows (Table 2m). When combining positive constant error and negative dynamic error, and vice versa (see Table 1g,h), *KGE* and *NSE* display better performance (Table 2g,h) than for single constant and dynamic error types (Table 2a-d).

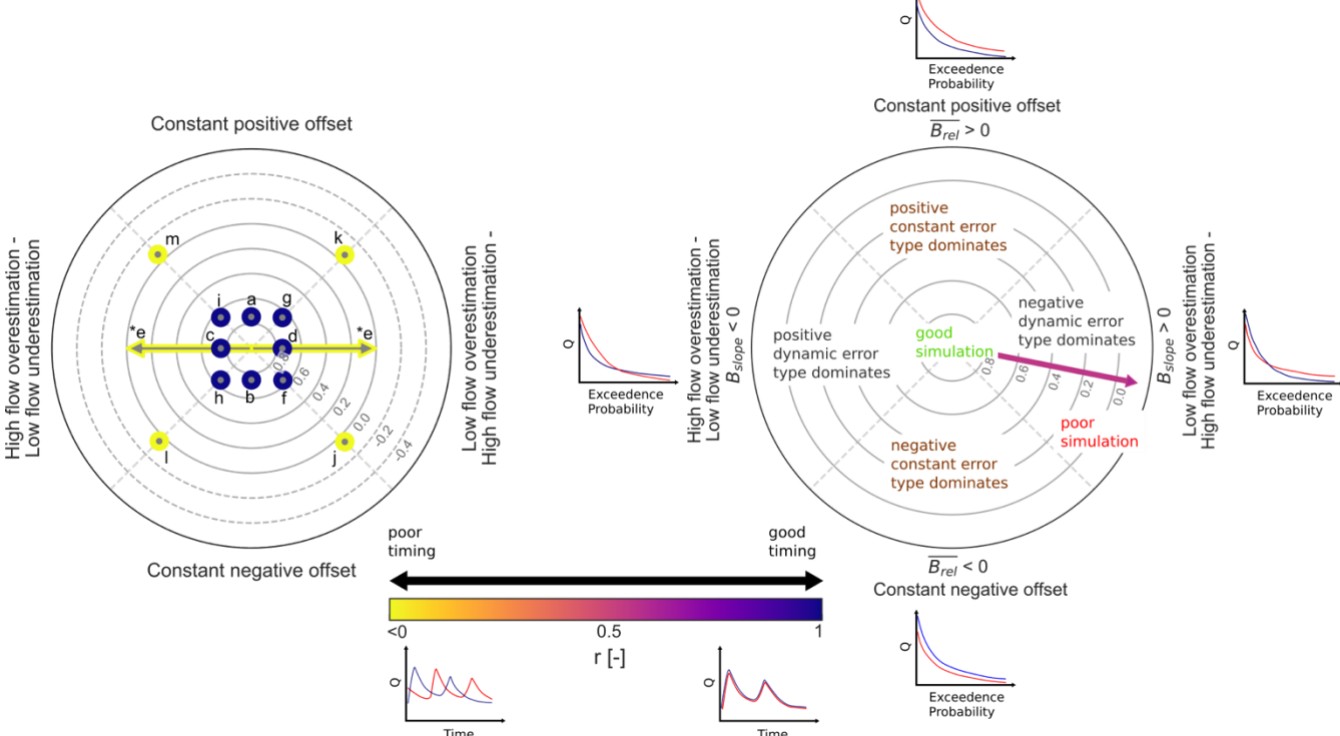

**Figure 3: (left) Diagnostic polar plot for manipulated time series generated by mimicking constant errors, dynamic errors and timing errors (a-m) visualizing the overall model performance (*DE*; contour lines) and contribution of constant error, dynamic error and timing error (purple (yellow) indicates temporal match (mismatch)). (e\*) timing error only: type of dynamic error cannot be distinguished. (right) Annotated diagnostic polar plot illustrating the interpretation (similar to Zipper et al. (2018)). Hypothetic FDC plots and hydrograph plots give examples for the error types.**

**Table 2: Comparison of *DE*, *KGE* and *NSE* calculated for manipulated time series generated by mimicking constant error, dynamic error and timing error (a-m). Lowest model performance for each error case is in bold.**

|     | a | b | c | d | e | f | g | h | i | j | k | l | m |
|-----|------|------|------|------|------|------|------|------|------|-------|-------|-------|-------|
| *DE* | 0.75 | 0.75 | 0.75 | 0.75 | 0 | 0.65 | **0.65** | **0.65** | 0.65 | -0.06 | -0.06 | -0.06 | -0.06 |
| *KGE* | **0.65** | **0.65** | **0.43** | **0.43** | 0 | **0.08** | 0.75 | 0.75 | **0.08** | **-0.36** | -0.04 | -0.04 | -0.36 |
| *NSE* | 0.9 | 0.9 | 0.7 | 0.7 | **-1** | 0.27 | 0.94 | 0.94 | 0.27 | -0.25 | **-0.59** | **-1.58** | **-3.26** |





## 3.2 Modelling example

In order to demonstrate the applicability, we also use simulated streamflow time series which have been derived from Addor
et al. (2017). Streamflow time series have been simulated by the coupled Snow-17 and SAC-SMA system for the same
catchment as in Fig. 1. We briefly summarize here their modelling approach consisting of Snow-17 which "is a conceptual air-
temperature-index snow accumulation and ablation model" (Newman et al., 2015) and SAC-SMA model which is "a
conceptual hydrologic model that includes representation of physical processes such as evapotranspiration, percolation, surface

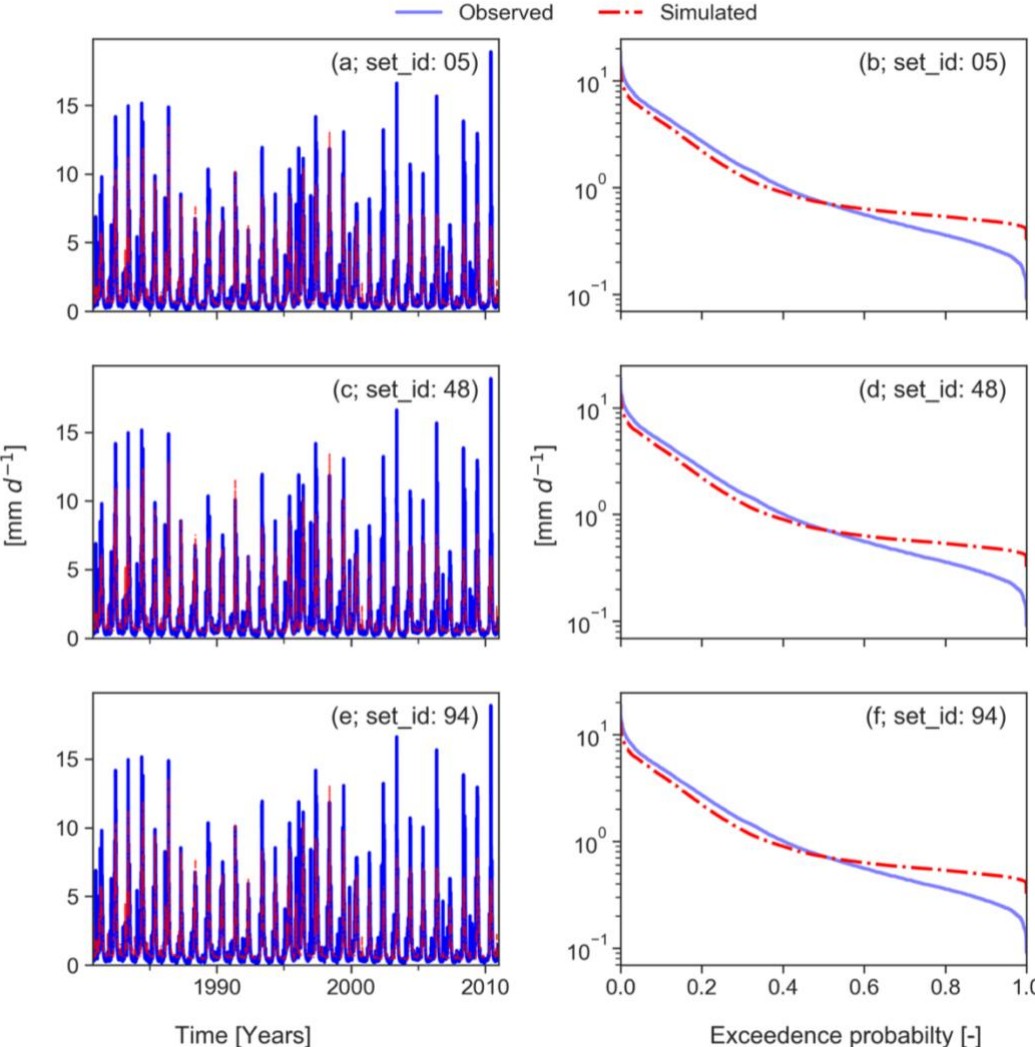

**Figure 4: Simulated and observed streamflow time series of modelling example (a, c and e) and the related flow duration curves (b,
d and f). Time series are derived from the CAMELS dataset (Addor et al., 2017). Observations and simulations belong to the same
catchment as in Figure 1. Simulations were produced by model runs with different parameter sets (set_id) but same input data (see
Newman et al., 2015).**



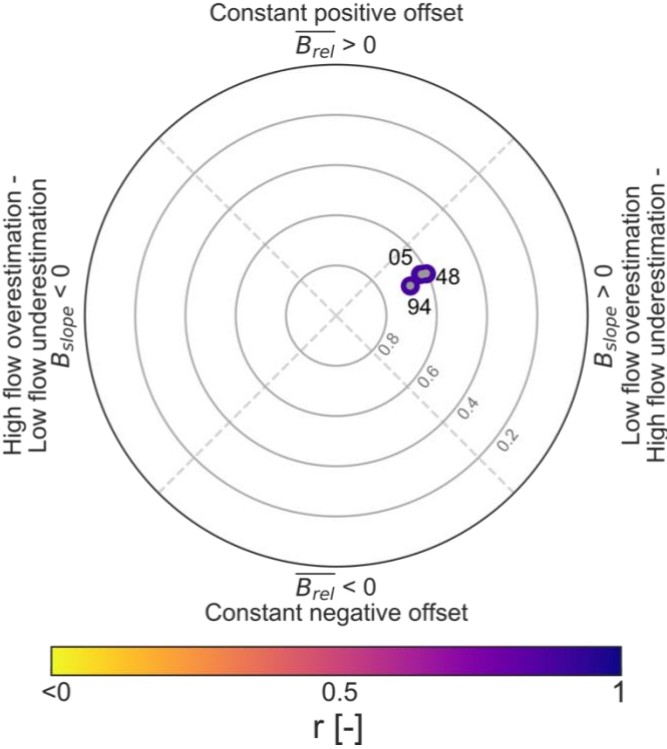

**Figure 5: Diagnostic polar plot for modelling example. Simulations were realised with three different parameter sets (05, 48, 94; see Fig. 4). All simulations perform well. However, the remaining error is dominated by a negative dynamic error type while timing is excellent.**

flow, and subsurface lateral flow" (Newman et al., 2015). Snow-17 runs first to partition precipitation into rain and snow and delivers the input for SAC-SMA model. For further details about the modelling procedure we refer to Sect. 3.1 in Newman et
al. (2015). In particular, we evaluated three model runs with different parameter sets, but the same input data. Simulated time series and simulated FDCs are shown in Fig. 4. The diagnostic polar plot for the three simulated time series is provided in Fig. 5. Simulations realised by parameter set with set_id 94 outperform the other two parameter sets. All simulations have in common, that positive dynamic error type (i.e. high flows are underestimated and low flows are overestimated) dominates accompanied by a slight positive constant error. Timing contributes least to the overall error.
After identifying the error types and its contributions, we can infer hints on how to improve the simulations. From a process-based (perceptual) perspective, the apparent negative dynamic error described by high flow underestimation and low flow overestimation suggest that process realism (e.g. snow melt, infiltration, storage outflow) appears to be deficient. Measures for improvement could start with adjusting the model parameters (e.g. refining the calibration procedure). If necessary, a follow-up measure could be to alter the model structure (e.g. adjusting the model equations). Additionally, there is a positive





constant error available. Because a constant error may be linked to input data errors, this implies that adjusting the input data
(e.g. precipitation correction, estimation of evapotranspiration) might improve the simulations.

## 4 Discussion

Aggregated performance metrics (e.g. *KGE* and *NSE*) are being criticised for not being hydrologically informative (Gupta et al., 2008). Although we systematically mimicked errors, we found an illogical pattern for *KGE* and *NSE* (Table 2) which makes

the interpretation of *KGE* and *NSE* more difficult. Particularly, in-depth analysis of the *KGE* metric terms revealed, that the $\beta$ term and $\alpha$ term are not orthogonal to each other (see Fig. S2 and Fig. S3c). We also lump model performance into a single value, but *DE* differs mainly in two points from the *KGE* and the *NSE*: (i) metric formulation is based on hydrological understanding instead of purely statistical understanding; (ii) diagnostic polar plots facilitate exploration of model deficiencies and diagnostics. When using *KGE* and *NSE* for evaluation purposes, we recommend a comparison to hydrologically

meaningful benchmarks which may add diagnostic value to *KGE* (e.g. Knoben et al., 2019) and *NSE* (e.g. Schaefli and Gupta, 2007). Based on such benchmark skill scores have been recently proposed to evaluate simulations (Knoben et al., 2019;Towner et al., 2019;Hirpa et al., 2018) to communicate model performance and to improve hydrologic interpretation. So far a way to define hydrologically meaningful benchmarks has not been extensively addressed by the hydrologic modelling community (Knoben et al., 2019).

Our approach focuses on model deficiencies. We do not propose a skill score measure for *DE* since skill scores introduce a scaling issue on communicating model errors (Knoben et al., 2019). *DE* does not rely on any benchmark to decide whether model diagnostics are required or not. Without considering any benchmark, *DE* may be interpreted as a deviation-from-perfect, measured by its constant error, dynamic and temporal error terms. In Sect. 2.2 (see Eq. 11) we introduced certain threshold for deviation-from-perfect (e.g. *DE*=0.91), if all error terms deviate by a certain degree (e.g. 5%; $\overline{B_{rel}}$=0.05, $|B_{area}|$=0.05, $r$=0.95).

Only for simulations in which deviation-from-perfect is sufficiently large, model diagnostics will be valuable.

By including FDC-based information into *DE*, we aimed for capturing rainfall-runoff response behaviour (Vogel and Fennessey, 1994) where different aspects of the FDC are inherently related to different processes (Ghotbi et al., 2020). But the way the dynamic error term is calculated (see Eqs. 4,5 and 7) limits the applicability to catchments with perennial streamflow. Measuring the timing error by linear correlation may also have limitations. Linear correlation can be criticised for neglecting

specific hydrological behaviour (Knoben et al., 2019), for example, flow recession or peak flow timing. But DE could also be calculated for different time periods and hence specific periods (e.g. wet periods versus dry periods) could be diagnosed separately.

Combining *DE* and diagnostic polar plots is, however, limited to three metric terms, because higher dimensional information cannot be effectively visualised by polar plots. We emphasize that the proposed metric terms of *DE* might not be perfectly

suitable for every evaluation purpose. For more specific evaluation, we suggest tailoring the proposed formulation of *DE* (see





Eq. 1) by exchanging the metric terms with, for example, low-flow-specific terms (e.g. see Fowler et al., 2018) or high-flow-specific terms (e.g. see Mizukami et al., 2019), respectively. Moreover, we suggest that different formulations of *DE* can be combined to a multi-criteria diagnostic evaluation (see Appendix).

**5 Conclusions**

The proposed approach is used as a tool for diagnostic model evaluation. Although errors may have multiple origins, these may be explored visually by diagnostic polar plots. A proof of concept and the application to a modelling example showed that errors coming from input data, model parameters and model structure can be unravelled. Particularly, diagnostic polar plots facilitate interpretation of model evaluation results. These plots may advance model development and application. The comparison to Kling-Gupta Efficiency and Nash-Sutcliffe Efficiency revealed, that they rely on a comparison to hydrological
meaningful benchmarks to become diagnostically interpretable. We tried to base the formulation of the newly introduced diagnostic efficiency is based on a general hydrological understanding and can thus be interpreted as deviation-from-perfect, we do not need to define benchmarks. More generally, our approach may serve as a blueprint for developing other Diagnostic Efficiency measures in the future.

*Code availability.* We provide a Python package *diag-eff* which can be used to calculate DE and the corresponding metric terms, produce diagnostic polar plots or mimic errors. The stable version can be installed via the Python Package Index (PyPI), and the current development version is available at https://github.com/schwemro/diag-eff.

*Data availability.* The observed and simulated streamflow time series are part of the open-source CAMELS dataset (Addor
et al., 2017). The data can be downloaded at https://ncar.github.io/hydrology/datasets/CAMELS_timeseries.

*Author contributions.* RS came up with initial thoughts. RS, DD and MW jointly developed and designed the methodology. RS developed the Python package, produced the figures and tables, and wrote the first draft of the manuscript. The manuscript was revised by DD and MW and edited by RS.


*Competing interests.* The authors declare that they have no conflict of interest.

*Acknowledgements.* We are grateful to Kerstin Stahl and Julia Dörrie for their comments on the language style and structure of the manuscript.


*Financial support.* This research has been supported by Helmholtz Association of German Research Centres (grant no. 42-2017).



## Appendix A

We briefly describe how *DE* could be extended to a tailored single-criteria metric (A1):

$$DE_{ext} = 1 - \sqrt{term_1^2 + term_2^2 + term_3^2}, \tag{A1}$$

Multiple single-criteria metric can be combined to a multi-criteria metric (A2):

$$DE_{multi-ext} = \frac{1}{N} \sum_{i=1}^{N} DE_{ext,i}, \tag{A2}$$

For a multi-criteria approach, diagnostic polar plots can be displayed for each single-criteria metric included into A2.

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
