# Peer review of "Technical note: Diagnostic efficiency – specific evaluation of model performance"

_Hydrology and Earth System Sciences, 2020_

## Referee Comment (RC1) · Anonymous Referee #1 · 8 Jul 2020

General comments

The technical note presents an intriguing new metric fusing together aspects of traditional efficiency and hydrologic signature metrics. The research is highly relevant to HESS, and the technical methodology is well described. The results used to demonstrate the utility of the new method of evaluating model performance are sufficient to support the conclusions of the manuscript. Overall, the material is well structured but there are some aspects which are unclear or insufficiently explained.

Specific comments

31: I do not see how traditional efficiency metrics only allow a binary choice between 'good' and 'poor'. They provide a gradation of relative performance. This should be

rephrased.

61: The justification is missing or misplaced. Why these three and not the other two?

62: The three types of model error are a key point in the manuscript, but this 'definition' is inadequate. Why these three types? What distinguishes the types? Listing potential sources of each type does not define anything. What is the difference between constant error from model parameters and dynamic error from model parameters?

71: Superficially, the DE metric looks like KGE (three component terms, covering bias, variability and correlation). The manuscript could be improved with an explicit contrast between the two, to highlight the novel aspects of the DE metric. Section 2.3 would be a good place, as it currently does not include a comparison, only formula regurgitation.

151: 'Mimicking' may not be the best term to describe the artificial errors generated for this demonstration. To mimic is to imitate, and the synthetic errors introduced to the observed time series are not intended to imitate anything in particular.

180: The summary table is very useful, but grid lines would improve the readability.

240: This paragraph has glossed over one key limitation of the new error metric. The 'negative dynamic error' lumps together high flow underestimation and low flow overestimation. The results presented in Figure 4 are a perfect example of why this is a limitation: all three time series have only low flow overestimation as a prominent error. How is the diagnostic polar plot (Fig 5) more informative than the FDC presented in Figure 4?

253: You have stated that the metric formulation is based on hydrological rather than purely statistical understanding, but this has not come out clearly earlier in the text. After all, one of your three component terms is identical to one used in the KGE. A more explicit justification for the hydrological basis would better support the novelty of your metric.

273: If the use of polar plots is limiting the information content, why not use some other

type of plot? For example, could a radar chart be used instead?

Technical corrections:

7: Should be 'part of' not 'part for'.

10: Unsatisfactory rather than unsatisfying.

10: Originate not origin.

15: Should be 'these three' not 'the three' as other error types are possible but not account for here.

21: Extra comma after 'suggests'.

31: Should this be "model performance using only a single numerical value"?

44: You do not need two qualifiers in this sentence, use either 'usually' or 'may only be' but not both.

52: This is not the best way to introduce the topic of model error or the stated topic of diagnostic efficiency.

55: 'Sources' may be more appropriate than 'origins' in this context.

96: The word 'does' is extraneous.

Figure 1: The figure could use a y-axis title, and I'm not sure that 'years' is an appropriate unit for dates.

149: Are the underscores appropriate for a caption?

152: In the following what? List, table or section?

285-287: Sentence contains grammatical errors, please correct.

---

## Referee Comment (RC2) · Anonymous Referee #2 · 1 Aug 2020

The authors present an interesting technical note in which they link the idea of diagnostic model evaluation with that of efficiency metrics. They propose a new metric in which they integrate terms to assess constant, dynamic and timing errors. I like the idea and the paper, but I am unclear about the way this metric and its terms are formulated, and how they relate to previous work. Hopefully my comments below help the authors to strengthen their argument.

MAJOR COMMENTS

[1] I understand that the first term of their metric is the relative bias of the FDC. Why is this a more hydrologically relevant and insightful term than other bias estimates? Can you show evidence for this claim?

[2] Similarly, I would find it more informative if the authors were to compare their terms to the terms in KGE and the non-parametric version by Pool et al. (2018) to really understand the differences. Why are these more informative and can it be shown?

[3] Would it not be more informative if the different parameter sets in Figure 4 were to show that different errors dominate? Why do they all show essentially identical FDCs? Maybe use more varied examples?

[4] Is the main problem one of aggregation? And hence loss of information. See for example the separate use of KGE terms in Gudmundsson et al. (2012). Even your second term is more informative because it leads to less aggregation and loss of information. Is this the key?

[5] It would be good if the authors would clarify their assumptions better and discuss how these might relate to reality. For example, they assume that precipitation has a consistent input data error. Some previous studies suggest that such an input error varies significantly between rainfall events (e.g. Yatheendradas et al., 2008, WRR). Similarly, for the other errors. It would strengthen the study significantly if the authors where to review the literature thoroughly for studies that discuss how these different errors manifest themselves (the authors lines 61ff). The three assumptions made here are key to the paper, but they are currently not supported by literature. I am not arguing that the authors' assumptions are wrong (though I might disagree partially), but they need to show evidence why these assumptions are reasonable. How to assign these errors is key here, but it is also something many people have argued about before.

[6] There have been others who raised the question of benchmarks before. For example Jan Seibert (https://eprints.ncl.ac.uk/file_store/production/246998/A084BCF1-F4EA-4EDF-AE6D-9E85C27A9DC4.pdf or Seibert, 2001). It would be good if the authors would review the literature more thoroughly on this topic.

[7] Section 3.7 is difficult to follow. Maybe this can easier be summarized in a figure? I find these error combinations difficult to read and compare. Maybe another figure

instead of the table?

REFERENCES

Gudmundsson, L., T. Wagener, L. M. Tallaksen, and K. Engeland (2012), Evaluation of nine large-scale hydrological models with respect to the seasonal runoff climatology in Europe, Water Resour. Res., 48, W11504, doi:10.1029/2011WR010911.

Pool, S., Vis, M., & Seibert, J. (2018). Evaluating model performance: towards a non-parametric variant of the Kling-Gupta efficiency. Hydrological Sciences Journal, 63(13-14), 1941-1953.

Seibert J. 2001. On the need for benchmarks in hydrological modelling. Hydrological Processes 15 (6): 1063–1064 DOI: 10.1002/hyp.446

Yatheendradas, S., T. Wagener, H. Gupta, C. Unkrich, D. Goodrich, M. Schaffner, and A. Stewart (2008), Understandinguncertainty in distributed flash flood forecasting for semiarid regions, Water Resour. Res., 44, W05S19, doi:10.1029/2007WR005940.

---

## Author Comment (AC1) · 28 Aug 2020

**Response to Reviewer #1**

We would like to thank the anonymous referee for his/her interest and the comments on our manuscript. Below, reviewer comments are in italic font and our replies are in plain blue font.

*General comments*

*The technical note presents an intriguing new metric fusing together aspects of traditional efficiency and hydrologic signature metrics. The research is highly relevant to HESS, and the technical methodology is well described. The results used to demonstrate the utility of the new method of evaluating model performance are sufficient to support the conclusions of the manuscript. Overall, the material is well structured but there are some aspects which are unclear or insufficiently explained.*

We thank the reviewer for his/her helpful comments.

*Specific comments*

*31: I do not see how traditional efficiency metrics only allow a binary choice between 'good' and 'poor'. They provide a gradation of relative performance. This should be rephrased.*

We fully agree and rephrase the text accordingly.

*61: The justification is missing or misplaced. Why these three and not the other two?*

We would like to point out that the justification is placed at line 55ff.

*62: The three types of model error are a key point in the manuscript, but this 'definition' is inadequate. Why these three types? What distinguishes the types? Listing potential sources of each type does not define anything. What is the difference between constant error from model parameters and dynamic error from model parameters?*

We used these three error types because constant, dynamic and timing errors are common model errors. We would like to emphasize, that each error type is calculated as an individual term in the DE. In order to assign the error types (constant, dynamic, timing) to error sources (input data error, parameters, model structure, etc.) contextual/expert knowledge (e.g. shortcomings of the input data) or statistical analysis (e.g. linking the error types with model parameters) is required. We will rephrase the definition and add further explanations.

*71: Superficially, the DE metric looks like KGE (three component terms, covering bias, variability and correlation). The manuscript could be improved with an explicit contrast between the two, to highlight the novel aspects of the DE metric. Section 2.3 would be a good place, as it currently does not include a comparison, only formula regurgitation.*

We will strengthen the difference and add a sentence including a comparison in Section 2.3. Furthermore, we would like to point out that the supplement contains a comparison between the *DE* terms and the *KGE* terms for the artificially generated errors (Figure S2, Figure S3 and Table S1) and for the modelling example (Figure S4 and Table S2). These results are discussed in Section 4 (see lines 248ff).

*151: 'Mimicking' may not be the best term to describe the artificial errors generated for this demonstration. To mimic is to imitate, and the synthetic errors introduced to the observed time series are not intended to imitate anything in particular.*

We agree and will rephrase the term mimicking into generation of artificial errors.

*180: The summary table is very useful, but grid lines would improve the readability.*

We add grid lines.

*240: This paragraph has glossed over one key limitation of the new error metric. The 'negative dynamic error' lumps together high flow underestimation and low flow overestimation. The results presented in Figure 4 are a perfect example of why this is a limitation: all three time series have only low flow overestimation as a prominent error. How is the diagnostic polar plot (Fig 5) more informative than the FDC presented in Figure 4?*

We agree that the lumping represents a limitation and we will add a paragraph to the manuscript. In most cases high flow underestimation and low flow overestimation are not equally prominent. We emphasize that with DE and the corresponding diagnostic polar plot only the main error can be identified. In order to explore more specific errors, we recommend to include specific signatures (see Appendix A).

A visual evaluation and comparison of the FDCs (see Figure 4) does not allow the identification of the best parameter set. For example, it would be difficult to find the "best" parameter set from 100 model runs just from the FDC. *KGE* and *NSE* do not provide any information on which parts of the FDC are underestimated or overestimated, respectively. Moreover, a separated interpretation of the FDC and the efficiency metric do not give any hint towards the error type. The strength of our approach is the combined visualization of the overall model performance and the different metric terms which enables the identification of the dominant error type. Figure 5 clearly shows which is the best parameter set and what are the dominant errors although the parameter sets perform slightly different.

*253: You have stated that the metric formulation is based on hydrological rather than purely statistical understanding, but this has not come out clearly earlier in the text. After all, one of your three component terms is identical to one used in the KGE. A more explicit justification for the hydrological basis would better support the novelty of your metric.*

Since the first two terms of DE are based on the FDC, we argue that this improves the hydrological understanding. We strengthen the hydrological justification in the manuscript. Moreover, we want to stress that the metric terms could be easily replaced with other hydrologic signatures (see Appendix A).

*273: If the use of polar plots is limiting the information content, why not use some other type of plot? For example, could a radar chart be used instead?*

The polar plot is just one way to visualize multidimensional information. Of course, radar chart could be used instead. The polar plot technique facilitates multiple evaluations (e.g. multiple simulations from different parameter sets or multiple simulations from different models) since points are used instead of polygon shapes.

*Technical corrections:*

*7: Should be 'part of' not 'part for'.*

*10: Unsatisfactory rather than unsatisfying.*

*10: Originate not origin.*

*15: Should be 'these three' not 'the three' as other error types are possible but not account for here.*

*21: Extra comma after 'suggests'.*

*31: Should this be "model performance using only a single numerical value"?*

*44: You do not need two qualifiers in this sentence, use either 'usually' or 'may only be' but not both.*

*52: This is not the best way to introduce the topic of model error or the stated topic of diagnostic efficiency.*

*55: 'Sources' may be more appropriate than 'origins' in this context.*

*96: The word 'does' is extraneous.*

*Figure 1: The figure could use a y-axis title, and I'm not sure that 'years' is an appropriate unit for dates.*

*149: Are the underscores appropriate for a caption?*

*152: In the following what? List, table or section?*

*285-287: Sentence contains grammatical errors, please correct.*

We will include all technical corrections in the manuscript.

---

## Author Comment (AC2) · 28 Aug 2020

**Response to Reviewer #2**

We would like to thank the anonymous referee for his/her interest and the comments on our manuscript. Below, reviewer comments are in italic font and our replies are in plain blue font.

*The authors present an interesting technical note in which they link the idea of diagnostic model evaluation with that of efficiency metrics. They propose a new metric in which they integrate terms to assess constant, dynamic and timing errors. I like the idea and the paper, but I am unclear about the way this metric and its terms are formulated, and how they relate to previous work. Hopefully my comments below help the authors to strengthen their argument.*
We thank the reviewer for his/her useful comments.

*MAJOR COMMENTS*
*[1] I understand that the first term of their metric is the relative bias of the FDC. Why is this a more hydrologically relevant and insightful term than other bias estimates? Can you show evidence for this claim?*
The relative bias of the FDC (i.e. constant bias) may have similar hydrological relevance than other bias estimates. Since we remove first the constant error (see Eq. 5) before we compute the dynamic error, we used the relative bias of the FDC for reasons of consistency.

*[2] Similarly, I would find it more informative if the authors were to compare their terms to the terms in KGE and the non-parametric version by Pool et al. (2018) to really understand the differences. Why are these more informative and can it be shown?*
For the comparison between the *DE* terms and the *KGE* terms for the artificially generated errors, we would like to refer to the supplement (Figure S2, Figure S3 and Table S1). Similarly, the comparison between the *DE* terms and the *KGE* terms for the modelling example, we would like to refer to the supplement (Figure S4 and Table S2). We will add a sentence to Section 2.3 (as suggested by reviewer #1) which strengthens the difference between DE terms and KGE terms. In addition, as already suggested in the paper, a non-parametric version of the DE could also be used (replacing Pearson's correlation coefficient with for example the Spearman's rank coefficient)

*[3] Would it not be more informative if the different parameter sets in Figure 4 were to show that different errors dominate? Why do they all show essentially identical FDCs? Maybe use more varied examples?*
The overall objective of the modelling example in Section 3.2 is to demonstrate the applicability of our approach. Of course we could have used an example for which different error dominates. In order to illustrate, when different errors dominate we would like to refer to Figure 2. The FDCs seem to be almost identical, because we compared three model runs which are among the ten best parameter sets. Figure 5 clearly shows which is the best parameter set and what are the dominant errors although the parameter sets perform slightly different.

*[4] Is the main problem one of aggregation? And hence loss of information. See for example the separate use of KGE terms in Gudmundsson et al. (2012). Even your second term is more informative because it leads to less aggregation and loss of information. Is this the key?*

We would like to point out, that we try to overcome the problem of aggregation by separating and visualising the model performance in combination with the metric terms. Using the polar plot technique, the results can be visualised in a disaggregated way. However, a certain level of aggregation cannot be avoided since each metric term already reflects an aggregation itself. We add a sentence to Section 4 which will highlight the value of including the metric terms into the model evaluation.

[5] It would be good if the authors would clarify their assumptions better and discuss how these might relate to reality. For example, they assume that precipitation has a consistent input data error. Some previous studies suggest that such an input error varies significantly between rainfall events (e.g. Yatheendradas et al., 2008, WRR). Similarly, for the other errors. It would strengthen the study significantly if the authors where to review the literature thoroughly for studies that discuss how these different errors manifest themselves (the authors lines 61ff). The three assumptions made here are key to the paper, but they are currently not supported by literature. I am not arguing that the authors' assumptions are wrong (though I might disagree partially), but they need to show evidence why these assumptions are reasonable. How to assign these errors is key here, but it is also something many people have argued about before.
We highly appreciate this critical comment. In order to assign the error sources contextual/expert knowledge (e.g. shortcomings of the input data) or statistical analysis (e.g. linking the error types with model parameters) is required. We will rephrase the definition, add further explanations and provide the missing references.

[6] There have been others who raised the question of benchmarks before. For example Jan Seibert (https://eprints.ncl.ac.uk/file_store/production/246998/A084BCF1-F4EA-4EDF-AE6D-9E85C27A9DC4.pdf or Seibert, 2001). It would be good if the authors would review the literature more thoroughly on this topic.
The point we want to make here is that DE does not require any benchmark for an improved hydrological interpretation (see lines 298ff).

[7] Section 3.7 is difficult to follow. Maybe this can easier be summarized in a figure? I find these error combinations difficult to read and compare. Maybe another figure instead of the table?
Unfortunately, there does not exist a Section 3.7. We assume that the comment addresses Section 3.1. We recommend using Figure 3 in combination with Table 1.

REFERENCES
Gudmundsson, L., T. Wagener, L. M. Tallaksen, and K. Engeland (2012), Evaluation of nine large-scale hydrological models with respect to the seasonal runoff climatology in Europe, Water Resour. Res., 48, W11504, doi:10.1029/2011WR010911.
Pool, S., Vis, M., & Seibert, J. (2018). Evaluating model performance: towards a non-parametric variant of the Kling-Gupta efficiency. Hydrological Sciences Journal, 63(13-14), 1941-1953.
Seibert J. 2001. On the need for benchmarks in hydrological modelling. Hydrological Processes 15 (6): 1063–1064 DOI: 10.1002/hyp.446

Yatheendradas, S., T. Wagener, H. Gupta, C. Unkrich, D. Goodrich, M. Schaffner, and A. Stewart (2008), Understanding uncertainty in distributed flash flood forecasting for semiarid regions, Water Resour. Res., 44, W05S19, doi:10.1029/2007WR005940.

---

## Referee Report (RR1)

Review of "Technical note: Diagnostic efficiency – specific evaluation of model performance" by Schwemmle et al.

**Summary**

The authors introduce a new efficiency metric called Diagnostic Efficiency (DE), which replaces the bias and variability components of the Kling-Gupta efficiency with two statistical measures derived from the observed and simulated flow duration curve. The goal is to improve the connection between efficiency metric score and specific errors in the modelling setup, so that diagnosing model deficiencies becomes easier. Several synthetic test cases are shown to describe the metric's functionality and the metric is also used on a real-world test case.

I have read this paper with interest and I think that metrics that help diagnose model failures are a relevant area of research. I do think that this paper needs some improvements before it can published. In my opinion, the benefits of DE are currently a bit overstated and there are some methodological choices that would benefit from a clearer explanation. In particular, (1) I do not think that the interpretation of DE's error types as being caused by specific types of model deficiencies is currently well-supported, and (2) I do think that DE needs to be used in combination with some form of hydrologically meaningful benchmark/threshold if it is to be used to determine if model simulations are deficient or not. Regarding the methodology, I have some questions about (1) how the constant error term can be interpreted in cases where the simulated FDC is both above and below the observed one; (2) how the calculation of Bdir works, and (3) how deviations between two FDCs can be used to trace model deficiencies. More details are in the comments below.

Kind regards,

Wouter Knoben

**Comments**

l11. "Input data"; it may be more accurate to rephrase this as "data uncertainty" because errors in the evaluation data could equally lead to unsatisfactory model performance although in that case it may be that what we consider as the "truth" is faulty and not the simulations.

l31. "value close to one indicates a better model performance". Given the nature of the paper, it would be good to define what is meant by "better model performance" and similar words and phrases. Some readers may interpret this as meaning that the model is an appropriate representation of the catchment in question (high model "fidelity"), but, as the authors indicate, a NSE or KGE score of 1 only indicates a perfect numerical match between observations and simulations (high model accuracy, but not necessarily for the right reasons). The metrics themselves do not provide any interpretations about how well the model simulations represent real-world hydrology and it would be good to be explicit about this.

L59. Do input data errors and observation uncertainty (l60) not fall under observations with insufficient accuracy mentioned in line 54?

L62. It's not immediately obvious to me why this paper addresses three out of the 5 error sources mentioned in lines 57-61. Can it be clarified why these three errors are the focus of this work?

L64. Using these three error terms seems the core assumption of this paper. The provided examples help in understanding what they mean but I think formal definitions of each error term should be included here.

L64. I would expect some form of justification to support the choice of these three types of errors. Are they sufficient to describe all possible deviations from observations that simulations could show?

L71. I expect the authors chose an equation of the form DE = 1-X to match the way NSE and KGE are formulated, but I think this makes the metric vulnerable to wrong interpretation. NSE is a skill score, where any simulations with NSE > 0 can be said to have outperformed the mean flow benchmark model included in the NSE equation. KGE has no such benchmark, but due to its formulation as KGE = 1-Y it is an easy mistake to assume that KGE = 0 has some distinct meaning even though it does not. DE does not seem to be a skill score either and I don't immediately see that DE = 0 has any special meaning. I would strongly recommend to reformulate the metric as DE = X, so that DE = 0 can be cleanly interpreted as "there are zero errors".

L77. It's not fully clear to me why $\overline{Brel}$ indicates a constant error. What happens in cases where the simulated FDC both overestimates and underestimates the observed one? $\overline{Brel}$ will likely still fall on one side of zero but that does not mean that all simulations showcase a constant error of $\overline{Brel}$.

L82. How would this equation deal with catchments where the observations drop to zero, but the simulations do not? This indicates a model error that should show during evaluation but equation 3 will break in such a scenario.

L94. The conceptual novelty of DE seems to be that it uses observed and simulated FDC's for two of its three metrics. For $\overline{Brel}$ and $|Barea|$, the series of observed and simulated Q values are thus not used as a time series but ordered into a flow duration curve. A major consequence of this is that the temporal connection between observations and simulations is mostly lost, because the two series are not compared on a per-timestep basis. With the data ordered as FDCs, it's no longer clear at which point in the simulation certain errors were generated and thus where model deficiencies may be found. An extreme example would be a case where a model matches all observations perfectly, but for some reason returns zero flow on those timesteps where the observations are highest. When the simulations are shown as a FDC, those zero flows will have exceedance probabilities of 100% and thus we might think that the model underestimates the low flows, even though the model in fact simulates the low flows just fine but massively underestimates the high flows. It is therefore unclear to me why quantifying the errors between both flow duration curves leads to increased understanding of model errors. From my point of view, it seems equally possible that the temporal disconnect between observations and simulations will mask certain model errors instead. I realize that neither NSE nor KGE would be of much use in this example either, but the almost complete temporal disconnect of the simulations and observations in DE is worthy of discussion. It could also be helpful to define a few more extreme cases of model errors and see to what extent DE can be used to trace those errors.

L102. I tried implementing equations 2, 3 and 5 with real data but could not reproduce 50% of the integral being positive values and 50% being negative. Instead, my Bres plot alternates between being

negative and positive and only ~45% of its values are negative (see figure). I have included my code below. Assuming that I didn't make any mistakes, can the authors clarify whether the equations and assumptions in the manuscript are correct?

[Figure]

**sort hydrographs to create FDC (scaling can be done in plot)**
qsim = np.asarray(sorted(sim))
qobs = np.asarray(sorted(obs))

**calculate B variables**
brel = (qsim-qobs)/qobs # Eq. 3
brel_bar = np.mean(brel) # Eq. 2
bres = brel - brel_bar # Eq. 5

L106. Why is this referred to as a slope? This equation seems to only change |Barea| back into Barea through a somewhat roundabout way. "Slope" implies some value with units [distance/distance].

L120. It's not entirely clear to me why |Brel_bar| has this specific threshold at 1.

L126. I understand that section 2.2 tries to outline different scenarios for Brel_bar, Bslope, DE and Del but I find this quite difficult to follow. I'm struggling to follow the reasoning that leads to equations 12-14 and feel a bit lost with all these variables that I'm seeing for the first time. Maybe a longer

explanation, or a graphical example, or placing the scenarios in a table or even a flowchart could help to clarify this section.

L166. "Note that the original temporal order is maintained." Is this correct? The FDC contains no temporal information. Should the mention of "FDC" on line 164 be "time series" instead? Also on line 168.

L191. "Interdependently… regions." I don't understand what this sentence means.

L202. "Numerically, … NSE." I suggest to remove this sentence. The fact that DE scores are higher than NSE and KGE scores is irrelevant (there is no reason why these scores can or should be compared in a relative sense) and referring to this as "better performance" may be confusing to readers who associate "better [model] performance" with "more accurate representations of real-world hydrology".

L207. "For example, lowest KGE values … (Table 2a-d)." I'm not sure how to interpret this sentence. Can this be clarified?

L237-243. I find this attribution of causes to certain error types very speculative. For example, underestimation of high flows and overestimation of low flows could equally indicate that precipitation input is smeared out over time, which tends to happen with gridded forcing products interpolated from station data, or with climate models that have a tendency to drizzle. Equally, a constant positive error (overestimation of flows) may indicate a model structure issue such as an inappropriate evaporation routine (not enough water returns to the atmosphere) or a "impervious runoff" routine that allows part of the incoming precipitation to bypass the soil moisture routine entirely or a "subsurface water exchange process" that imports water from an underlying aquifer. Parameter issues could also play a role here, for example if soil moisture storage capacity is set too low and part of the incoming precipitation directly goes into streamflow as saturation excess runoff, or if evaporation is limited by some form of inappropriately set wilting point. I suggest to either remove this section or better support why certain types of errors must (or are at least most likely to) be generated from the causes described here.

L261. I think point (iii) is a somewhat optimistic view. The link between error type and associated model deficiencies is a bit tenuous in the current manuscript (see previous comment) and needs to be better supported before this can be presented as a feature provided by DE.

L269. DE may not use a benchmark simulation, but it does have the same issue that it is difficult to say which DE scores indicate that a model is "good enough". The authors have not justified their use of a 5% deviation threshold on each of the DE components, which I assume was chosen for illustrative purposes only. Therefore, DE has the same interpretation challenges as KGE (which is also a deviation-from-perfect kind of metric) and the recommendations for KGE should apply to DE too. I suggest to clarify this in the text.

L291. "A proof of concept and the application to a modelling example showed that errors coming from input data, model parameters and model structure can be unravelled with the help of expert knowledge or a statistical analysis. Particularly, diagnostic polar plots facilitate interpretation of model evaluation results. These plots may advance model development and application." This seems to be mostly speculation in the current manuscript (see comment about lines 237-243). I suggest to either improve the support for this statement or remove it from the conclusions.

L294. "We tried to base the formulation of the newly introduced diagnostic efficiency on a general hydrological understanding and can thus be interpreted as deviation-from-perfect, we do not need to define benchmarks." This seems a bit optimistic. DE cannot answer the question "is my model good enough" without a statement about the level of deviation-from-perfect that is considered acceptable for a given purpose. Justifying where this level is set is functionally equivalent to specifying a benchmark. I suggest to remove the last part of this sentence.

L319. I appreciate this stepping stone to more work on efficiency metrics but the provided equations seem a bit trivial and in the case of A1 perhaps even overly specific. There is no real need for future metrics to be of the shape De = 1 – X (arguably, DE = X would lead to less ambiguity) nor do such metrics need to have three components and not two or four or some other number. I expect that this appendix can be removed without harming the main manuscript.

**Editorial**

l29. Replace "can be measured by only" with "with".

l34. "satisfying" > "satisfactory"

l160, l162. "requires" > "required"?

---

## Referee Report (RR2)

Review 2 of "Technical note: Diagnostic efficiency – specific evaluation of model performance" by Schwemmle et al

**Summary**

I believe the authors have responded well to most review comments. They have made significant changes to the manuscript. The metric has been rewritten in the form DE = X, which in my opinion is a lot less vulnerable to misinterpretation, and the connection between error values and error sources, which in my and other reviewers' opinions was somewhat speculative, has been removed.

There are a few points that I think deserve a bit more attention. I have added my own responses to those provided by the authors in a new document and uploaded that as a reviewer attachment. Line numbers refer to those in the track-changes manuscript. Author comments are kept in blue. Most importantly, I think the manuscript would be strengthened if the authors would add a cautionary discussion note about potential pitfalls in interpreting the mismatch between a simulated and observed flow duration curve and the associated $Bre\overline{l}$ value. More explanation below.

I hadn't mentioned this in the last review, but making the code to create the polar plots publicly available is a great idea and very helpful.

Kind regards,

Wouter Knoben

**General comments**
< majority of text removed for brevity >
For example, in case the high flow part of the simulated FDC is greater than the high flow part of the observed FDC indicates that the simulations overestimate the peak flows. < …>

I don't think this is quite correct. To me, stating that simulations overestimate peak flows implies that the simulations give us peaks at the same time as the observations have peaks, but that the simulations are too high. I would say that because the observed and simulated hydrographs are sorted independently and contain no temporal information, all we can say from such a case is that the distribution of simulated flows contains higher flows than observed but we cannot conclude when these model failures occur.

Two helpful thought experiments may be (1) to take any hydrograph and create synthetic simulations by increasing the low flows of this hydrograph until they are higher than the highest observed flows, and (2) to take a strongly seasonal hydrograph and create synthetic simulations by shifting this hydrograph in time until the "simulations" overestimate the low flows and underestimate the peaks. In the first case, the simulated FDC has higher flows than the observed one, but stating that the simulations overestimate the peaks would be incorrect (they overestimate the low flows). In the second case, the model overestimates the lows and underestimates the peaks but the FDCs for both hydrographs are identical, meaning that there is no clear 1:1 connection between model error and FDC mismatch and hence care is needed to interpret FDC (mis)match. Hopefully this clarifies why I think that conclusions such as the one in blue above cannot be derived from comparing two FDCs. I would strongly encourage the authors to reflect this

line of reasoning in their manuscript and update the text where appropriate. See also comment about L94 below.

**Specific comments**
*L82. How would this equation deal with catchments where the observations drop to zero, but the simulations do not? This indicates a model error that should show during evaluation but equation 3 will break in such a scenario.*
We are fully aware about this shortcoming. Therefore, the metric is only valid for catchments with perennial streamflow (see line 268).

It may be good to make this mention slightly more prominent, perhaps by moving it directly below Eq. 3. In my experience discussion items get overlooked more easily and this is a fairly critical aspect of the proposed metric (i.e. it being only applicable to regions with perennial flow) that deserves visibility.

*L94. The conceptual novelty of DE seems to be that it uses observed and simulated FDC's for two of its three metrics. For $\overline{B_{rel}}$ and $|B_{area}|$, the series of observed and simulated Q values are thus not used as a time series but ordered into a flow duration curve. A major consequence of this is that the temporal connection between observations and simulations is mostly lost, because the two series are not compared on a per-time step basis. With the data ordered as FDCs, it's no longer clear at which point in the simulation certain errors were generated and thus where model deficiencies may be found. An extreme example would be a case where a model matches all observations perfectly, but for some reason returns zero flow on those time steps where the observations are highest. When the simulations are shown as a FDC, those zero flows will have exceedance probabilities of 100% and thus we might think that the model underestimates the low flows, even though the model in fact simulates the low flows just fine but massively underestimates the high flows. It is therefore unclear to me why quantifying the errors between both flow duration curves leads to increased understanding of model errors. From my point of view, it seems equally possible that the temporal disconnect between observations and simulations will mask certain model errors instead. I realize that neither NSE nor KGE would be of much use in this example either, but the almost complete temporal disconnect of the simulations and observations in DE is worthy of discussion. It could also be helpful to define a few more extreme cases of model errors and see to what extent DE can be used to trace those errors.*
Again the metric is only valid for catchments with perennial streamflow (see line 268). Of course, comparing the observed FDC and simulated FDC disconnects the time steps. However, the timing error term is not related to the FDC. We used Pearson's correlation coefficient (see Eq. 6) to compare the simulated time series and observed time series on per-time step basis. We want to emphasize that *DE* is not the perfect metric. Instead *DE* represents an alternative tool which can be used in addition for model evaluation.

I agree with this response that the timing error is captured as part of the Pearson correlation. What I hoped but failed to accurately convey with this comment is to urge caution with statements that do assign a temporal component to conclusions based on just comparing both FDCs. See also the general comment above. This is important in e.g. lines 255-257: "All simulations have in common, that positive dynamic error type (i.e. high flows are underestimated and low flows are overestimated) dominates accompanied by a slight positive constant error. Timing contributes least to the overall error." It is very clear in Figure 4 (FDC column) that all three models simulate a narrower range of flows than observed, by simulating lower high flows and higher low flows. Equally in Figure 4 (hydrograph column) it can be seen that, if we arbitrarily classify the peak between months 4 and 8 as "high flows" and the remainder as "low flows", the model both underestimates (months 2-4) and overestimates (months 8-1) the low flows, instead of only overestimating the low flows as the text indicates. This particular aspect of model failure cannot be deduced by just looking at the FDC and $\overline{B_{rel}}$ values.

I would again strongly encourage the authors to add a discussion paragraph where this issue is discussed. I think that the need to be very careful about interpreting FDC and $\overline{B_{rel}}$ results is something that can easily be missed if the manuscript does not devote sufficient attention to it.

*L102. I tried implementing equations 2, 3 and 5 with real data but could not reproduce 50% of the integral being positive values and 50% being negative. Instead, my Bres plot alternates between being negative and positive and only ~45% of its values are negative (see figure). I have included my code below. Assuming that I didn't make any mistakes, can the authors clarify whether the equations and assumptions in the manuscript are correct?*

The equations are correct. We do not assume that 50% of the be either entirely positive or entirely negative. In your case (i.e. ~45% of its values are negative) it means the left part of the FDC is mostly underestimated and the right part of the FDC is mostly overestimated.

I'm glad I implemented the equations correctly. I based this comment on line 111: "Since we removed the constant error (see Eq. 5), the left half of the integral is positive and the right half (i.e. $50_{th}$ percentile to $100_{th}$ percentile) will, thus, be negative and vice versa if the left half of the integral is negative." I suggest to update/remove this sentence.

*L120. It's not entirely clear to me why |Brel_bar| has this specific threshold at 1.*
The threshold refers to the letter *l*.

My mistake, thanks for clarifying.

**Editorial**
L122. "we introduce certain threshold" > "we introduce a certain threshold"

---

## Author Response (AR2)

**Response to Reviewer #1**

We would like to thank the anonymous referee for his/her interest and the comments on our manuscript. Below, reviewer comments are in italic font and our replies are in plain blue font.

**General comments**

The manuscript presents a potential useful new metric and method for evaluating hydrologic model inadequacy, which is definitely in the journal scope. The technical note has been somewhat improved by the revisions, and most comments have been addressed to some degree. The methods are valid and fully outlined, the results are adequately discussed and sufficient to support the conclusions. The overall structure of the technical note is fine, but there are some issues with the text and figures which detract from the comprehensibility of the paper. Some minor revisions should be completed prior to publication. We thank the reviewer for his/her helpful comments.

**Specific comments**

53-62: This section is not clear. The text states that observations should be checked for accuracy, but then uncertainties in observations remain, therefore we will not focus on uncertainties in observations. This is not helpful in clarifying why some sources of uncertainty were the focus of the new metric. We rephrased the section and replaced observations with evaluation data.

65-70: This may be a minor point, but what good is it three defined error types to be linked to potential error sources when, in your example, any of the three types can be linked to the same source (precipitation input)? If expert knowledge is need, how is the novel metric helping?

The novel metric shows which of three error types is dominating. The expert can then define which error may dominate and provide measure to either reduce the error or at least better consider the effect of the uncertainty.

98: Polar plots are not very much like clocks, given both angle and distance from the origin matter in polar plots. This comparison is more confusing than helpful.

We agree and removed the comparison.

*Figure 1: The y-axis has no title.* We added a title to the y-axis.

*Figure 2: The y-axis has no title.* We added a title to the y-axis.

**Figure 4: Why are the FDC y-axes in log space? It initially appears as though the simulation only over estimates low flows, when the magnitude of high flow under estimation is in fact similar.**

Using a linear y-axis the differences for the low flows would have been barely visible, therefore we decided to use log discharge axis. We are aware that this emphasis the low flow part more, but there is unfortunately no really good approach in comparing high flow and low flow with a similar scale.

Figure 5: One benefit of the novel metric and method which may not be highlighted in the manuscript is its ability to facilitate simulation comparisons. While the FDC curves contain more information than the polar plot, a single

diagnostic polar plot can clearly summarize the errors in dozens or hundreds of simulations (which would be unreadable in a single FDC plot).

We included a sentence which highlights this benefit. Thanks for pointing this out, this was clearly in our mind, but sometimes things get lost at the end.

Technical corrections:

30: 'Measure' twice

191: 'Interdependently' is either being used incorrectly, or was not the word intended

256: 'Illogical' is not a good descriptor for the quirks and irregularities of the NSE and KGE

We included all technical corrections in the manuscript.

The authors may also wish to review the erratic comma usage, although this is not impeding the manuscript's comprehensibility.

We reviewed the erratic comma usage.

**Response to Reviewer #3**

We would like to thank Wouter Knoben for his interest and the comments on our manuscript. Below, reviewer comments are in italic font and our replies are in plain blue font. We would also like to point out, that some remarks of this new reviewer are more difficult to approach, since changes we did in the first round of this review process were not known or considered or may have even be in a different direction as the reviews in the first round.

**General comments**

The authors introduce a new efficiency metric called Diagnostic Efficiency (DE), which replaces the bias and variability components of the Kling-Gupta efficiency with two statistical measures derived from the observed and simulated flow duration curve. The goal is to improve the connection between efficiency metric score and specific errors in the modelling setup, so that diagnosing model deficiencies becomes easier. Several synthetic test cases are shown to describe the metric's functionality and the metric is also used on a real-world test case.

I have read this paper with interest and I think that metrics that help diagnose model failures are a relevant area of research. I do think that this paper needs some improvements before it can published. In my opinion, the benefits of DE are currently a bit overstated and there are some methodological choices that would benefit from a clearer explanation. In particular, (1) I do not think that the interpretation of DE's error types as being caused by specific types of model deficiencies is currently well-supported, and (2) I do think that DE needs to be used in combination with some form of hydrologically meaningful benchmark/threshold if it is to be used to determine if model simulations are deficient or not. Regarding the methodology, I have some questions about (1) how the constant error term can be interpreted in cases where the simulated FDC is both above and below the observed one; (2) how the calculation of Bdir works, and (3) how deviations between two FDCs can be used to trace model deficiencies. More details are in the comments below.

We thank the reviewer for his helpful comments. The provided support on the causes of the specific error types may serve as a general advice how the errors might be linked to their source. This is more thought as an example, certainly not as a strong linkage to specific model deficiencies. We want to emphasize that our goal is not to replace current metrics. Moreover, *DE* is not designed to answer the question whether simulations are "good enough". This is in our opinion a philosophical question and many others have already tried to answer this question. But this is a problem for all metrics except there is a perfect agreement. Instead, we aim to provide a new alternative metric which focuses on the error identification. Moreover, we aim to facilitate the comparison between multiple simulations.

The constant error term represents the average of the relative bias distribution. Positive values for the constant error term can be always interpreted as overestimation on average whereas negative values can be interpreted as underestimation on average.  $B_{dir}$  is the integral of residual error (i.e. relative bias of the two FDCs after subtracting the constant error (see Eq. 5)) that reaches from 0th percentile to the 50th percentile. The deviation between the observed FDC and the simulated FDC contains some hydrologic information. For example, in case the high flow part of the simulated FDC is greater than the high flow part of the observed FDC indicates that the simulations overestimate the peak flows.  $B_{dir}$  is required to calculate  $B_{slope}$ .

Specific comments

111. "Input data"; it may be more accurate to rephrase this as "data uncertainty" because errors in the evaluation data could equally lead to unsatisfactory model performance although in that case it may be that what we consider as the "truth" is faulty and not the simulations.

We assume that the evaluation data represents the truth. We do not want to question whether the evaluation data is truly representative or not. Instead, we presume that the data quality of the evaluation data is checked beforehand. Answering the question about the "truth" is partly a philosophical one and cannot be answered by this technical note.

131. "value close to one indicates a better model performance". Given the nature of the paper, it would be good to define what is meant by "better model performance" and similar words and phrases. Some readers may interpret this as meaning that the model is an appropriate representation of the catchment in question (high model "fidelity"), but, as the authors indicate, a NSE or KGE score of 1 only indicates a perfect numerical match between observations and simulations (high model accuracy, but not necessarily for the right reasons). The metrics themselves do not provide any interpretations about how well the model simulations represent real-world hydrology and it would be good to be explicit about this.

We agree and include the term accuracy to describe the model performance.

L59. Do input data errors and observation uncertainty (160) not fall under observations with insufficient accuracy mentioned in line 54?

With observations we mean the data used for the evaluation. We rephrase this line.

**L62. It's not immediately obvious to me why this paper addresses three out of the 5 error sources mentioned in lines 57-61. Can it be clarified why these three errors are the focus of this work?**

In general, metrics are based on a comparison between observations and simulations. Consequently, metrics do not have the purpose to quantify the uncertainty of the evaluation data. Accounting for errors caused by initial/boundary conditions require a more complex approach. The approach we presented here is not suitable to quantify errors caused by initial and/or boundary conditions.

L64. Using these three error terms seems the core assumption of this paper. The provided examples help in understanding what they mean but I think formal definitions of each error term should be included here. We added some formal definitions. And we would also like to point out, that the core of the paper is the new metric

and its potential advantages to other metrics, not the error terms.

*L64. I would expect some form of justification to support the choice of these three types of errors. Are they sufficient to describe all possible deviations from observations that simulations could show?*

These three error types are common errors generated by many hydrological models. The three error types might not be sufficient to describe all possible deviations. In order to account for a wider range of deviations, we recommend to use different error terms or choosing a multi-criteria approach (see line 277f).

L71. I expect the authors chose an equation of the form DE = 1-X to match the way NSE and KGE are formulated, but I think this makes the metric vulnerable to wrong interpretation. NSE is a skill score, where any simulations with NSE > 0 can be said to have outperformed the mean flow benchmark model included in the NSE equation. KGE has no such benchmark, but due to its formulation as KGE = 1-Y it is an easy mistake to assume that KGE = 0 has some distinct meaning even though it does not. DE does not seem to be a skill score either and I don't immediately see that DE = 0 has any special meaning. I would strongly recommend to reformulate the metric as DE = X, so that DE = 0can be cleanly interpreted as "there are zero errors".

We agree on using an error score and a perfect fit should indicate a value of 0. The idea of using DE = 1-Y was to make it comparable with KGE. So it may also be discussed, why KGE was accepted the way it is. But we see the point and change DE from DE = 1-Y to DE = X = 0 means zero errors.

**L77. It's not fully clear to me why $Bre\overline{\overline{t}}$ indicates a constant $e\overline{\overline{t}}$ or. What happens in cases where the simulated FDC both overestimates and underestimates the observed one? $Br\overline{\overline{t}}$ will likely still fall on one side of zero but that does not mean that all simulations showcase a constant error of $Bre\overline{\overline{t}}$ .**

 $B_{rel}$  represents the average of the relative bias distribution. Positive values for the constant error term can be always interpreted as an average overestimation whereas negative values can be always interpreted as an average underestimation. Overestimation and underestimation of the simulated FDC are captured by  $B_{area}$ .  $B_{rel}$  describes only the average behaviour.

*L82. How would this equation deal with catchments where the observations drop to zero, but the simulations do not? This indicates a model error that should show during evaluation but equation 3 will break in such a scenario.*

We are fully aware about this shortcoming. Therefore, the metric is only valid for catchments with perennial streamflow (see line 268).

L94. The  $\frac{1}{2}$  inceptual novelty of DE seems to be that it uses observed and simulated FDC's for two of its three metrics. For Bret and Bared, the series of observed and simulated Q values are thus not used as a time series but ordered into a flow duration curve. A major consequence of this is that the temporal connection between observations and simulations is mostly lost, because the two series are not compared on a per-time step basis. With the data ordered as FDCs, it's no longer clear at which point in the simulation certain errors were generated and thus where model deficiencies may be found. An extreme example would be a case where a model matches all observations perfectly, but for some reason returns zero flow on those time steps where the observations are highest. When the simulations are shown as a FDC, those zero flows will have exceedance probabilities of 100% and thus we might think that the model underestimates the low flows, even though the model in fact simulates the low flows just fine but massively underestimates the high flows. It is therefore unclear to me why quantifying the errors between both flow duration curves leads to increased understanding of model errors. From my point of view, it seems equally possible that the temporal disconnect between observations and simulations will mask certain model errors instead. I realize that neither NSE nor KGE would be of much use in this example either, but the almost complete temporal disconnect of the simulations and observations in DE is worthy of discussion. It could also be helpful to define a few more extreme cases of model errors and see to what extent DE can be used to trace those errors.

Again the metric is only valid for catchments with perennial streamflow (see line 268). Of course, comparing the observed FDC and simulated FDC disconnects the time steps. However, the timing error term is not related to the FDC. We used Pearson's correlation coefficient (see Eq. 6) to compare the simulated time series and observed time series on per-time step basis. We want to emphasize that DE is not the perfect metric. Instead DE represents an alternative tool which can be used in addition for model evaluation.

L102. I tried implementing equations 2, 3 and 5 with real data but could not reproduce 50% of the integral being positive values and 50% being negative. Instead, my Bres plot alternates between being negative and positive and only  $\sim$ 45% of its values are negative (see figure). I have included my code below. Assuming that I didn't make any mistakes, can the authors clarify whether the equations and assumptions in the manuscript are correct?

The equations are correct. We do not assume that 50% of the be either entirely positive or entirely negative. In your case (i.e.  $\sim$ 45% of its values are negative) it means the left part of the FDC is mostly underestimated and the right part of the FDC is mostly overestimated.

L106. Why is this referred to as a slope? This equation seems to only change |Barea| back into Barea through a somewhat roundabout way. "Slope" implies some value with units [distance/distance].

We use the term slope which is referred to the inclination of the dynamic error. For example, Yilmaz et al. (2008) used a similar terminology. In case  $B_{slope}$  is positive low flows are underestimated and high flows are overestimated (i.e. left integral of  $B_{res}$  is negative and right integral  $B_{res}$  is positive). Vice versa, if  $B_{slope}$  is negative low flows are overestimated and high flows are underestimated (i.e. left integral of  $B_{res}$  is positive and right integral  $B_{res}$  is negative).

*L120. It's not entirely clear to me why* |*Brel\_bar*| *has this specific threshold at 1.* The threshold refers to the letter *l*.

L126. I understand that section 2.2 tries to outline different scenarios for Brel\_bar, Bslope, DE and Del but I find this quite difficult to follow. I'm struggling to follow the reasoning that leads to equations 12-14 and feel a bit lost with all these variables that I'm seeing for the first time. Maybe a longer explanation, or a graphical example, or placing the scenarios in a table or even a flowchart could help to clarify this section.

There is only one new variable which is  $DE_l$ . The equations 12-14 represent the conditions for which a diagnosis can be made.

L166. "Note that the original temporal order is maintained." Is this correct? The FDC contains no temporal information. Should the mention of "FDC" on line 164 be "time series" instead? Also on line 168.

The sentence was corrected. It should be time series.

*L191. "Interdependently... regions." I don't understand what this sentence means.* We rephrased the sentence.

L202. "Numerically, ... NSE." I suggest to remove this sentence. The fact that DE scores are higher than NSE and KGE scores is irrelevant (there is no reason why these scores can or should be compared in a relative sense) and referring to this as "better performance" may be confusing to readers who associate "better [model] performance" with "more accurate representations of real-world hydrology". We removed the sentence.

L207. "For example, lowest KGE values ... (Table 2a-d)." I'm not sure how to interpret this sentence. Can this be clarified?

We rephrased the sentence. The point we want to make here is that *KGE* and *NSE* are differently sensitive to the generated errors.

L237-243. I find this attribution of causes to certain error types very speculative. For example, underestimation of high flows and overestimation of low flows could equally indicate that precipitation input is smeared out over time, which tends to happen with gridded forcing products interpolated from station data, or with climate models that have a tendency to drizzle. Equally, a constant positive error (overestimation of flows) may indicate a model structure issue such as an inappropriate evaporation routine (not enough water returns to the atmosphere) or a "impervious runoff" routine that allows part of the incoming precipitation to bypass the soil moisture routine entirely or a "subsurface water exchange process" that imports water from an underlying aquifer. Parameter issues could also play a role here, for example if soil moisture storage capacity is set too low and part of the incoming precipitation directly goes into streamflow as saturation excess runoff, or if evaporation is limited by some form of inappropriately set wilting point. I suggest to either remove this section or better support why certain types of errors must (or are at least most likely to) be generated from the causes described here.

We removed this pargraph.

L261. I think point (iii) is a somewhat optimistic view. The link between error type and associated model deficiencies is a bit tenuous in the current manuscript (see previous comment) and needs to be better supported before this can be presented as a feature provided by DE. We agree and rephrased point (iii).

L269. DE may not use a benchmark simulation, but it does have the same issue that it is difficult to say which DE scores indicate that a model is "good enough". The authors have not justified their use of a 5% deviation threshold on each of the DE components, which I assume was chosen for illustrative purposes only. Therefore, DE has the same

interpretation challenges as KGE (which is also a deviation-from- perfect kind of metric) and the recommendations for KGE should apply to DE too. I suggest to clarify this in the text.

We have chosen the 5% deviation mainly for illustrative purposes. The threshold can be set to any value. This is more about whether it is worth to diagnose the error and not about whether a model is "good enough". In comparison to *KGE*, the deviation is measured by a percentage of error.

L291. "A proof of concept and the application to a modelling example showed that errors coming from input data, model parameters and model structure can be unravelled with the help of expert knowledge or a statistical analysis. Particularly, diagnostic polar plots facilitate interpretation of model evaluation results. These plots may advance model development and application." This seems to be mostly speculation in the current manuscript (see comment about lines 237-243). I suggest to either improve the support for this statement or remove it from the conclusions. We rephrased the sentence and removed the speculation.

L294. "We tried to base the formulation of the newly introduced diagnostic efficiency on a general hydrological understanding and can thus be interpreted as deviation-from-perfect, we do not need to define benchmarks." This seems a bit optimistic. DE cannot answer the question "is my model good enough" without a statement about the level of deviation-from-perfect that is considered acceptable for a given purpose. Justifying where this level is set is functionally equivalent to specifying a benchmark. I suggest to remove the last part of this sentence.

We removed the last part of the sentence. We want to emphasize that our goal is not to answer "Is my model good enough?". The goal is to facilitate the comparison of multiple simulations and to easily identify dominant error types. In addition, we provide a graphical tool (i.e. diagnostic polar plots).

L319. I appreciate this stepping stone to more work on efficiency metrics but the provided equations seem a bit trivial and in the case of A1 perhaps even overly specific. There is no real need for future metrics to be of the shape De = 1-X (arguably, DE = X would lead to less ambiguity) nor do such metrics need to have three components and not two or four or some other number. I expect that this appendix can be removed without harming the main manuscript. We appreciate this comment, but we decided to keep the Appendix in the manuscript.

**Editorial**

129. Replace "can be measured by only" with "with".
134. "satisfying" > "satisfactory"
1160, 1162. "requires" > "required"?
We included all technical corrections in the manuscript.

**List of all relevant changes**

- We added formal definitions for the error terms
- We changed the formula of DE from DE=I-Y to DE=X
- We further strengthened the difference between *DE* and *KGE*

[revised manuscript text omitted]
 = \frac{1 - \sqrt{B_{rel}^2 + |B_{area}|^2 + (r-1)^2}}{\sqrt{B_{rel}^2} + |B_{area}|^2 + (r-1)^2},$$
(1)

where Brel is a measure for the constant error, |Barea| for the dynamic error, and r for the timing error. Similar to NSE and KGE,
DE ranges from +0 to -∞.∞ and DE = +0 indicates, that there are no errors (i.e. perfect agreement between simulations and observations-). In contrast to KGE and NSE, DE represents an error score. This means, that model performance is decreasing for increasing values of DE.

First, we introduce the three terms which define the *DE*. The first two terms  $\overline{B_{rel}}$  and  $|B_{area}|$  are based on the flow duration curve (FDC). Since FDC-based signatures do not include information on temporal performance, we have added correlation (*r*)

85 between the simulated time series and the observed time series as a third term.  $\overline{B_{rel}}$  reflects the constant error and is represented by the arithmetic mean of the relative bias (Eq. 2):

$$\overline{B_{rel}} = \frac{1}{N} \sum_{i=0}^{i=1} B_{rel}(i),$$
(2)

*i* represents the exceedance probability, N the total number of data points and  $B_{rel}$  is the relative bias of the simulated and observed flow duration curve;  $\overline{B_{rel}} = 0$  indicates no constant error;  $\overline{B_{rel}} < 0$  indicates a negative bias;  $\overline{B_{rel}} > 0$  indicates a positive bias. The relative bias between the simulated and observed flow duration curve  $(B_{rel})$  calculates as follows (Eq. 3):

90

$$B_{rel}(\mathbf{i}) = \frac{Q_{sim}(\mathbf{i}) - Q_{obs}(\mathbf{i})}{Q_{obs}(\mathbf{i})},\tag{3}$$

 $Q_{sim}$  is the simulated streamflow at exceedance probability *i* and  $Q_{obs}$  the observed streamflow at exceedance probability *i*. The dynamic error is described by the absolute area of the residual bias ( $|B_{area}|$ ; Eq. 4):

$$|B_{area}| = \int_0^1 |B_{res}(i)| \ di,$$
(4)

where the residual bias  $B_{res}$  is integrated over the entire domain of the flow duration curve. Combining Eq. (2) and Eq. (3) 95 results in:

$$B_{res}(i) = B_{rel}(i) - \overline{B_{rel}},$$
(5)

by subtracting  $\overline{B_{rel}}$  we remove the constant error and the dynamic error remains.  $|B_{area}| = 0$  indicates no dynamic error;  $|B_{area}|$ > 0 indicates a dynamic error.

100 To consider timing errors, the Pearson's correlation coefficient (r) is calculated (Eq. 6):

$$r = \frac{\sum_{l=1}^{n} (Q_{obs}(l) - \mu_{obs})(Q_{sim}(l) - \mu_{sim})}{\sqrt{(\sum_{l=1}^{n} (Q_{obs}(l) - \mu_{obs})^2)(\sum_{l=1}^{n} (Q_{sim}(l) - \mu_{sim})^2)}} r = \frac{\sum_{l=1}^{n} (Q_{obs}(l) - \mu_{obs})(Q_{sim}(l) - \mu_{sim})}{\sqrt{(\sum_{l=1}^{n} (Q_{obs}(l) - \mu_{obs})^2)(\sum_{l=1}^{n} (Q_{sim}(l) - \mu_{sim})^2)}},$$
(6)

where  $Q_{sim}$  is the simulated streamflow at time t,  $Q_{obs}$  the observed streamflow at time t,  $\mu_{obs}$  the simulated mean streamflow, and  $\mu_{obs}$  the observed mean streamflow. Other non-parametric correlation measures could be used as well.

**2.2 Diagnostic polar plot**

DE can be used as another aggregated efficiency by simply calculating the overall model performance error. However, the 105 aggregated value only allows for a limited diagnosis since information of the metric terms is not interpreted. Thus, we project DE and its metric terms in a radial plane (i.e. similar to a clock) to construct a diagnostic polar plot. An annotated version for a diagnostic polar plot is given in Fig. 3. For the diagnostic polar plot, we calculate the direction of the dynamic error  $(B_{dir};$ Eq. 7):

110
$$B_{dir} = \int_0^{0.5} B_{res}(i) \, di,$$
 (7)

where the integral of  $B_{res}$  includes values from 0th percentile to 50th percentile. Since we removed the constant error (see Eq. 5), the left half of the integral is positive and the right half (i.e. 50th percentile to 100th percentile) will, thus, be negative and vice versa if the left half of the integral is negative.

In order to differentiate the dynamic error type, we computed the slope of the residual bias ( $B_{slope}$ ; Eq. 8):

115
$$B_{slope} = \begin{cases} |B_{area}| \cdot (-1), & B_{dir} > 0 \\ |B_{area}| & , & B_{dir} < 0 , \\ 0 & , & B_{dir} = 0 \end{cases}$$
 (8)

 $B_{slope} = 0$  expresses no dynamic error;  $B_{slope} < 0$  indicates that there is a tendency of simulations to overestimate high flows and/or underestimate low flows while  $B_{slope} > 0$  indicates a tendency of simulations to underestimate high flows and/or overestimate low flows.

We used the inverse tangent to derive the ratio between constant error and dynamic error in radians ( $\varphi$ , Eq. 9):

120
$$\varphi = \arctan 2(\overline{B_{rel}}, B_{slope}),$$
 (9)

Instead of using a benchmark to decide whether model diagnostics is valuable or not, we introduce certain threshold for deviation-from-perfect. We set a threshold value (l) for which metric terms deviate from perfect and insert it in Eq. (1):

$$DE_{l} = \frac{1 - \sqrt{l^{2} + l^{2} + ((1 - l) - 1)^{2}}}{\sqrt{l^{2} + l^{2} + ((1 - l) - 1)^{2}}},$$
(11)

for this study *l* is set by default to 0.05. Here, we assume that for a deficient simulation each metric term deviates at least 5% from its best value. *l* can be either relaxed or expanded depending on the requirements of model accuracy. Correspondingly,  $DE_l$  represents a threshold which discerns between a deficient simulation ( $DE \le DE_t$ ) and a good simulation ( $DE > DE_t$ ).to discern whether an error diagnosis ( $DE > DE_t$ ) is valuable.

Finally, the following conditions describe whether a diagnosis can be drawn (Eq. 12):

$$\quad Diagnosis = \begin{cases} \frac{yes,}{|B_{rel}| \le 1\&B_{slope} > 1\&DE \le DE_t} \\ \frac{yes,}{|B_{rel}| > 1\&B_{slope} \le 1\&DE \le DE_t} \\ \frac{yes,}{|B_{rel}| > 1\&B_{slope} \ge 1\&DE \le DE_t} \\ \frac{yes,}{|B_{rel}| > 1\&B_{slope} > 1\&DE \le DE_t} \end{cases} \begin{cases} yes, & |\overline{B}_{rel}| \le 1\&B_{slope} > 1\&DE > DE_t \\ yes, & |\overline{B}_{rel}| > 1\&B_{slope} \ge 1\&DE > DE_t \\ \frac{yes,}{|B_{rel}| > 1\&B_{slope} > 1\&DE \le DE_t} \end{cases} \end{cases}$$
(12)

There exists a special case for which timing error only can be diagnosed (Eq. 13):

 $Diagnosis = timing \ error \ only, \qquad |\overline{B_{rel}}| \le \frac{1 \& B_{slope}}{1 \& DE} \le \frac{1 \& DE}{2} \le 1 \& DE > DE_{l_{*}}$ (13)

If *DE* and its metric terms are within the boundaries of acceptance, no diagnosis is required which is expressed by the following conditions (Eq. 14):

$$Diagnosis = no, \qquad |\overline{B_{rel}}| \le 1 \& B_{slope} \le 1 \& DE \rightarrow DE_{t} \le DE_{l}, \tag{14}$$

In this case, the model performance is sufficiently accurate and can be denoted as a good simulation errors are too small.

**2.3 Comparison to KGE and NSE**

135

In order to allow a comparison to commonly used *KGE* and *NSE*, we calculated the overall metric values and for *KGE* its three individual metric terms. We used the original *KGE* proposed by Gupta et al. (2009):

$$KGE = 1 - \sqrt{(\beta - 1)^2 + (\alpha - 1)^2 + (r - 1)^2},$$
(15)

where  $\beta$  is the bias error,  $\alpha$  represents the flow variability error, and *r* shows the linear correlation between simulations and observations (Eq. 16):

$$KGE = 1 - \sqrt{\left(\frac{\mu_{sim}}{\mu_{obs}} - 1\right)^2 + \left(\frac{\sigma_{sim}}{\sigma_{obs}} - 1\right)^2 + (r - 1)^2},$$
(16)

145 where σobs is the standard deviation in observations, σsim the standard deviation in simulations. Moreover, we applied the polar plot concept (see Sect. 2.2) to *KGE* and the accompanying three metric terms. In contrast to *DE* (see Sect. 2.1) the formulation of *KGE* is entirely based on statistical signatures. By replacing the first two terms of *KGE* with FDC based signatures, we aim to improve the hydrological focus and provide a stronger link to the error sources. 2.1), KGE ranges from 1 to -∞ and the metric formulation of *KGE* is entirely based on statistical signatures. By replacing the first two terms of *KGE* with FDC based signatures, we aim to improve the hydrological focus and provide a stronger link to the error sources. 2.1), KGE ranges from 1 to -∞ and the metric formulation of *KGE* is entirely based on statistical signatures. By replacing the first two terms of *KGE* with FDC-based signatures, we aim to improve the hydrological focus and provide a stronger link to hydrological processes (e.g. Ghotbi et al., 2020).

NSE (Nash and Sutcliffe, 1970) calculates as follows (Eq. 17):

$$NSE = 1 - \frac{\sum_{t=1}^{t=T} (Q_{obs}(t) - Q_{sim}(t))^2}{\sum_{t=1}^{t=T} (Q_{obs}(t) - \mu_{obs})^2},$$
(17)

where *T* is the total number of time steps,  $Q_{sim}$  the simulated streamflow at time *t*,  $Q_{obs}$  the observed streamflow at time *t* and 155  $\mu_{obs}$ . NSE = 1 displays perfect fit between simulations and observations; NSE = 0 indicates that simulations performs equally well as the mean of the observations; NSE < 0 indicates that simulations perform worse than the mean of the observations.

**3 Proof of concept**

To provide a proof of concept any perennial streamflow time series coming from a near-natural catchment and having sufficiently long temporal record (i.e. > 30 years) may be used. We selected an observed streamflow time series from the CAMELS dataset (Fig. 1; Addor et al., 2017). In order to generate specific model errors, we systematically manipulated the observed time series. Thus, we produced different time series which serve as a surrogate for simulated time series with a certain error type which we call manipulated time series. These manipulated time series are characterised by a single error type or multiple error types, respectively. We calculated *DE* for each manipulated time series and visualised the results in a diagnostic polar plot.

---

## Author Response (AR3)

**List of all relevant changes**

- We added the error contribution of high flows and low flows and display the contribution in the diagnostic polar plots

**2nd Response to Reviewer #3**

We would like to thank Wouter Knoben for his interest and the comments on our manuscript. Below, reviewer comments are in italic font and our replies are in plain blue font.

**Summary**

I believe the authors have responded well to most review comments. They have made significant changes to the manuscript. The metric has been rewritten in the form DE = X, which in my opinion is a lot less vulnerable to misinterpretation, and the connection between error values and error sources, which in my and other reviewers' opinions was somewhat speculative, has been removed.

There are a few points that I think deserve a bit more attention. I have added my own responses to those provided by the authors in a new document and uploaded that as a reviewer attachment. Line numbers refer to those in the track-changes manuscript. Author comments are kept in blue. Most importantly, I think the manuscript would be strengthened if the authors would add a cautionary discussion note about potential pitfalls in interpreting the mismatch between a simulated and observed flow duration curve and the associated  $Bret \overline{f}$  value. More explanation below.

I hadn't mentioned this in the last review, but making the code to create the polar plots publicly available is a great idea and very helpful.

Kind regards, Wouter Knoben

**General comments**

< majority of text removed for brevity >

For example, in case the high flow part of the simulated FDC is greater than the high flow part of the observed FDC indicates that the simulations overestimate the peak flows. < ... >

I don't think this is quite correct. To me, stating that simulations overestimate peak flows implies that the simulations give us peaks at the same time as the observations have peaks, but that the simulations are too high. I would say that because the observed and simulated hydrographs are sorted independently and contain no temporal information, all we can say from such a case is that the distribution of simulated flows contains higher flows than observed but we cannot conclude when these model failures occur.

Two helpful thought experiments may be (1) to take any hydrograph and create synthetic simulations by increasing the low flows of this hydrograph until they are higher than the highest observed flows, and (2) to take a strongly seasonal hydrograph and create synthetic simulations by shifting this hydrograph in time until the "simulations" overestimate the low flows and underestimate the peaks. In the first case, the simulated FDC has higher flows than the observed one, but stating that the simulations overestimate the peaks would be incorrect (they overestimate the low flows). In the second case, the model overestimates the lows and underestimates the peaks but the FDCs for both hydrographs are identical, meaning that there is no clear 1:1 connection between model error and FDC mismatch and hence care is needed to interpret FDC (mis)match. Hopefully this clarifies why I think that conclusions such as the one in blue above cannot be derived from comparing two FDCs. I would strongly encourage the authors to reflect this line of reasoning in their manuscript and update the text where

**appropriate. See also comment about L94 below.**

Thought experiment (1) leads to a positive constant error and a negative dynamic error (i.e. low flow overestimation and high flow underestimation). This is because the dynamic error is calculated over the entire flow domain (see Eq. (4)). However, in such case high flows are neither underestimated nor overestimated (i.e. the dynamic error can be fully described by "low flow" errors). We adjusted the equations (see Eq. 7-13). We calculate the error contribution of high flows and low flows and include them in the diagnostic polar plots.

In case of thought experiment (2) the two FDCs would not indicate any error. However, in such case a large timing error will be indicated. A large timing error may be a clear hint that simulations require a more detailed evaluation.

**Specific comments**

L82. How would this equation deal with catchments where the observations drop to zero, but the simulations do not? This indicates a model error that should show during evaluation but equation 3 will break in such a scenario.

We are fully aware about this shortcoming. Therefore, the metric is only valid for catchments with perennial streamflow (see line 268).

It may be good to make this mention slightly more prominent, perhaps by moving it directly below Eq. 3. In my experience discussion items get overlooked more easily and this is a fairly critical aspect of the proposed metric (i.e. it being only applicable to regions with perennial flow) that deserves visibility.

**We added the missing information to Eq. (3).**

L94. The conceptual novelty of DE seems to be that it uses observed and simulated FDC's for two of its three metrics. For  $Bre\bar{t}$  and |Barea|, the series of observed and simulated Q values are thus not used as a time series but ordered into a flow duration curve. A major consequence of this is that the temporal connection between observations and simulations is mostly lost, because the two series are not compared on a per-time step basis. With the data ordered as FDCs, it's no longer clear at which point in the simulation certain errors were generated and thus where model deficiencies may be found. An extreme example would be a case where a model matches all observations perfectly, but for some reason returns zero flow on those time steps where the observations are highest. When the simulations are shown as a FDC, those zero flows will have exceedance probabilities of 100% and thus we might think that the model underestimates the low flows, even though the model in fact simulates the low flows just fine but massively underestimates the high flows. It is therefore unclear to me why quantifying the errors between both flow duration curves leads to increased understanding of model errors. From my point of view, it seems equally possible that the temporal disconnect between observations and simulations will mask certain model errors instead. I realize that neither NSE nor KGE would be of much use in this example either, but the almost complete temporal disconnect of the simulations and observations in DE is worthy of discussion. It could also be helpful to define a few more extreme cases of model errors and see to what extent DE can be used to trace those errors.

Again the metric is only valid for catchments with perennial streamflow (see line 268). Of course, comparing the observed FDC and simulated FDC disconnects the time steps. However, the timing error term is not related to the FDC. We used Pearson's correlation coefficient (see Eq. 6) to compare the simulated time series and observed time series on per-time step basis. We want to emphasize that *DE* is not the perfect metric. Instead *DE* represents an alternative tool which can be used in addition for model evaluation.

I agree with this response that the timing error is captured as part of the Pearson correlation. What I hoped but failed to accurately convey with this comment is to urge caution with statements that do assign a temporal component to conclusions based on just comparing both FDCs. See also the general comment above. This is important in e.g. lines 255-257: "All simulations have in common, that positive dynamic error type (i.e. high flows are underestimated and low flows are overestimated) dominates accompanied by a slight positive constant error. Timing contributes least to the overall error." It is very clear in Figure 4 (FDC column) that all three models simulate a narrower range of flows than observed, by simulating lower high flows and higher low flows. Equally in Figure 4 (hydrograph column) it can be seen that, if we arbitrarily classify the peak between months 4 and 8 as "high flows" and the remainder as "low flows", the model both underestimates (months 2-4) and overestimates (months 8-1) the low flows, instead of only overestimating the low flows as the text indicates. This particular aspect of model failure cannot be deduced by just looking at the FDC and Bret values.

Again we are fully aware that our approach is not perfect. To overcome this shortcoming, we recommend to further specify the proposed metric formulation (see lines 441-445 and Appendix A). However, in Figure 4 we would argue that our approach is correct. Observed discharge rates below  $\sim 1 \text{ mm/day}$  (i.e. "low flow" part of the FDC) are overestimated whereas observed discharge rates above  $\sim 1 \text{ mm/day}$  (i.e. "high flow" part of the FDC) are underestimated. This is clearly visible in Figure 4. Therefore, months 2-4 are related to the "high flow" part since discharge rates are greater than 1 mm/day.

I would again strongly encourage the authors to add a discussion para  $\frac{1}{2}$  approximate approxima

Since we refined our approach by adding the error contribution of high flows and low flows, the results can be cleanly interpreted. We amended the text accordingly. This should be sufficient and a discussion paragraph may not be necessary.

L102. I tried implementing equations 2, 3 and 5 with real data but could not reproduce 50% of the integral being positive values and 50% being negative. Instead, my Bres plot alternates between being negative and positive and only  $\sim$ 45% of its values are negative (see figure). I have included my code below. Assuming that I didn't make any mistakes, can the authors clarify whether the equations and assumptions in the manuscript are correct?

The equations are correct. We do not assume that 50% to be either entirely positive or entirely negative. In your case (i.e.  $\sim$ 45% of its values are negative) it means the left part of the FDC is mostly underestimated and the right part of the FDC is mostly overestimated.

I'm glad I implemented the equations correctly. I based this comment on line 111: "Since we removed the constant error (see Eq. 5), the left half of the integral is positive and the right half (i.e. 50th percentile to 100th percentile) will, thus, be negative and vice versa if the left half of the integral is negative." I suggest to update/remove this sentence.

**We updated the sentence.**

L120. It's not entirely clear to me why |Brel\_bar| has this specific threshold at 1.

The threshold refers to the letter *l*.

My mistake, thanks for clarifying.

**Editorial**

*L122.* "we introduce certain threshold" > "we introduce a certain threshold"

We corrected the sentence.